# EXPLORING SYNTHESIZABLE CHEMICAL SPACE WITH ITERATIVE PATHWAY REFINEMENTS

**Seul Lee**[*]
KAIST
seul.lee@kaist.ac.kr

**Karsten Kreis & Srimukh Prasad Veccham & Meng Liu & Danny Reidenbach &
Saee Paliwal & Weili Nie[†] & Arash Vahdat[†]**
NVIDIA
{kkreis,sveccham,menliu,dreidenbach,saeep,wnie,avahdat}@nvidia.com

## ABSTRACT

A well-known pitfall of molecular generative models is that they are not guaranteed to generate synthesizable molecules. Existing solutions for this problem often struggle to effectively navigate exponentially large combinatorial space of synthesizable molecules and suffer from poor coverage. To address this problem, we introduce ReaSyn, an iterative generative pathway refinement framework that obtains synthesizable analogs to input molecules by projecting them onto synthesizable space. Specifically, we propose a simple synthetic pathway representation that allows for generating pathways in both bottom-up and top-down traversal of synthetic trees. We design ReaSyn so that both bottom-up and top-down pathways can be sampled with a single unified autoregressive model. ReaSyn can thus iteratively refine subtrees of generated synthetic trees in a bidirectional manner. Further, we introduce a discrete flow model that refines the generated pathway at the entire pathway level with edit operations: insertion, deletion, and substitution. The iterative refinement cycle of (1) bottom-up decoding, (2) top-down decoding, and (3) holistic editing constitutes a powerful pathway reasoning strategy, allowing the model to explore the vast space of synthesizable molecules. Experimentally, ReaSyn achieves the highest reconstruction rate and pathway diversity in synthesizable molecule reconstruction and the highest optimization performance in synthesizable goal-directed molecular optimization, and significantly outperforms previous synthesizable projection methods in synthesizable hit expansion. These results highlight ReaSyn's superior ability to navigate combinatorially-large synthesizable chemical space. Our code is available at https://github.com/NVIDIA-Digital-Bio/ReaSyn.

## 1 INTRODUCTION

Discovering molecules with desired properties in the chemical space comprises the core of drug discovery. However, drug discovery pipelines are challenging and labor-intensive due to their vast design space and multi-objective nature. Molecular generative models have recently emerged as a notable breakthrough with the potential to greatly accelerate the drug discovery process. However, their practical impact has remained limited as they often suffer from a common shortcoming: generated drug candidates are often synthetically inaccessible (Segler et al., 2018b; Gao & Coley, 2020; Walters & Barzilay, 2021).

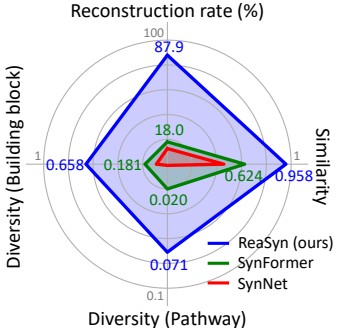

Figure 1: **Synthesizable molecule reconstruction** on test molecules constructed from ZINC250k. Full results are provided in Table 1.

---

[*]Work during an internship at NVIDIA.
[†]Equal advising.

Figure 2: **(a) Bottom-up and top-down traversal of a synthetic tree. (b) Overall framework of ReaSyn.** ReaSyn's generation cycle consists of three steps. First, an initial synthetic tree is generated by the autoregressive model in a bottom-up direction. Next, the autoregressive model repredicts a randomly selected subtree in a top-down direction. Finally, the Edit Flow model refines the generated tree in a holistic manner. This process can be repeated multiple times, and the best pathway that yields a product molecule of the highest similarity to the given target molecule is selected as the final solution. The sampling processes of the autoregressive model (the first and the second steps) and the Edit Bridge model are depicted in Figure 3(a) and Figure 3(b), respectively.

This problem arises from neglecting synthesizability during multi-objective molecular optimization, which often leads to exploration outside the synthesizable space. Although this issue can be tackled by incorporating a synthesizability score as an additional optimization objective (Ertl & Schuffenhauer, 2009; Coley et al., 2018; Thakkar et al., 2021), this approach is largely impractical since the heuristic scores are often poor proxies. This is because synthesizability is a complex function of molecular structure, and existing proxies cannot take available building block stocks or reaction selectivity into account (Gao & Coley, 2020).

An alternative solution is to constrain the design space to synthetically accessible space by generating synthetic pathways instead of solely generating final product molecules. This can be obtained either by generating synthesizable molecules in a *de novo* way (Bradshaw et al., 2020; Swanson et al., 2024; Koziarski et al., 2024; Cretu et al., 2025; Seo et al., 2025) or by projecting (possibly unsynthesizable) input molecules into the synthesizable space (Gao et al., 2021; Luo et al., 2024; Gao et al., 2025; Sun et al., 2025). In the latter approach, often called synthesizable projection or analog generation, a model learns to correct unsynthesizable molecules by generating pathways that lead to synthesizable analogs with similar structures. In contrast to the former approach, synthesizable projection benefits from a versatile and modular design and can be used with any off-the-shelf molecule generation method. It can also be applied to explore neighborhoods in the synthesizable space, allowing the model to perform different molecule *optimization tasks*, including hit expansion or lead optimization. Given these benefits, in this paper, we aim to solve the synthesizable projection problem using generative models.

When generating synthetic pathways in an autoregressive setting, a key decision is the direction in which the tree-structured pathway is generated (Figure 2(a)). Bottom-up generation (Bradshaw et al., 2020; Gao et al., 2021; Swanson et al., 2024; Luo et al., 2024; Gao et al., 2025; Koziarski et al., 2024; Cretu et al., 2025; Seo et al., 2025) and top-down generation (Sun et al., 2025) each have distinct advantages: the former can start from valid building blocks, and the latter aligns with the chemist's intuition. In this paper, we argue that identifying synthetic pathways for a given molecule is a *search* problem that benefits from unifying the two approaches: a model must navigate an exponentially large combinatorial space of possible intermediates and reactions, reasoning over sequences of chemical transformations that yield analogs. To this end, we introduce ReaSyn[1], a unified framework that can generate synthetic pathways both bottom-up and top-down (Figure 2(b) and Figure 3). This bidirectional design allows us to initialize a set of synthetic pathways given an input molecule and iteratively refine them in both directions. This iteration is required as changes at a node must be propagated both upward in the tree (to update the remaining synthetic pathway compatible with the new intermediate product) and downward in the tree (to regenerate a subtree that would yield the updated product).

The upward or downward sampling given a synthetic tree can be considered as edit operations that refine trees partially in one direction. To take these refinement operations one step further, we also introduce a tree editing scheme based on Edit Flow (Havasi et al., 2025) which we term as *Edit Bridge* (Figure 3(b)). Specifically, Edit Bridge is a novel discrete flow over pathway sequences that further refines the given pathway generated by the autoregressive model through edit operations: insertion, deletion, and substitution. This in turn allows ReaSyn to jointly edit both the tree skeleton

---

[1]Inspired by the reasoning nature of our problem, ReaSyn is pronounced as "reason".

Figure 3: ReaSyn adopts an encoder-decoder Transformer architecture. After the encoder encodes the input molecule, the decoder predicts the synthetic pathways of its synthesizable analogs. `[START]` and `[END]` tokens are omitted for simplicity. **(a) Bidirectional synthetic pathway generation of ReaSyn.** ReaSyn's autoregressive model predicts the synthetic pathways in the sequential representation. ReaSyn's training and inference scheme tailored for the bidirectional synthetic pathway generation enables to designate a specific sampling direction using a single autoregressive model. **(b) Holistic pathway editing of ReaSyn.** ReaSyn's Edit Bridge model takes the full pathway generated by the autoregressive model and jointly edits the tree skeleton and semantics.

and semantics at the entire pathway level. To the best of our knowledge, this paper is the first to bridge between learned and data distributions via editing operations.

Generative modeling over synthetic pathways requires choosing a pathway representation. Prior works (Bradshaw et al., 2020; Gao et al., 2021; Swanson et al., 2024; Luo et al., 2024; Gao et al., 2025; Koziarski et al., 2024; Cretu et al., 2025; Seo et al., 2025; Sun et al., 2025) have used hierarchical representation, including node type (i.e., reaction vs. building block) and node features (i.e., reaction class or building block features like Morgan fingerprints (Morgan, 1965). Striving for simplicity, ReaSyn introduces a simple sequential representation for traversing synthetic trees. ReaSyn eliminates (1) information loss in molecular fingerprints by directly using SMILES to represent building blocks and (2) error accumulation and architectural complexity arising from the hierarchical representations. Using this representation, ReaSyn adopts a novel training and inference scheme that can enforce a specific traversal direction in a single encoder-decoder Transformer (Vaswani et al., 2017).

We experimentally validate the effectiveness and versatility of ReaSyn on various tasks including synthesizable molecule reconstruction, synthesizable goal-directed molecular optimization, and synthesizable hit expansion. ReaSyn outperforms existing methods with superior reconstruction rates and pathway diversity in synthesizable molecule reconstruction, and achieves state-of-the-art performance in finding synthesizable chemical optima in synthesizable goal-directed molecular optimization and hit expansion. For example, as shown in Figure 1, ReaSyn significantly outperforms existing methods in exploring the synthetic chemical space to reconstruct synthesizable molecules (87.9% vs. 18.0% reconstruction rate) with greater sampling diversity (0.658 vs. 0.181 building block diversity). All these results indicate that ReaSyn has broader coverage of the synthesizable chemical space, highlighting its efficacy as a practical tool in real-world drug discovery scenarios. We summarize our contributions as follows:

- We introduce a simple synthetic pathway representation that enables traversal of synthesis trees in both bottom-up and top-down directions.

- We propose a new bidirectional search strategy in synthetic pathway space that unifies bottom-up and top-down sampling.

- We propose Edit Bridge, a novel discrete flow that bridges between the learned distribution in a generative model and the data distribution through edit operations.

- We propose ReaSyn, a framework that integrates bottom-up decoding, top-down decoding, and holistic editing to establish a multi-view pathway generation method for synthesizable projection.

- We demonstrate the effectiveness and versatility of ReaSyn in various synthesizable molecule generation and optimization tasks through extensive experiments.

## 2 RELATED WORK

**Synthesizable molecule design.** Molecular synthesizability is a vital problem in drug discovery and therefore has received a lot of attention. In synthesizable molecule design, the synthesizable chemical space is defined by sets of reaction rules and building blocks, and the design space is constrained to this

space. Various algorithms have been proposed to navigate the space to find synthesizable drug candidates, including autoencoders (AEs) (Bradshaw et al., 2020), variational autoencoders (VAEs) (Pedawi et al., 2022), Bayesian optimization (Korovina et al., 2020), genetic algorithms (GAs) (Gao et al., 2021), Monte Carlo tree search (MCTS) (Swanson et al., 2024), and GFlowNets (Koziarski et al., 2024; Seo et al., 2025; Cretu et al., 2025). However, most of them are either only capable of considering single-step synthetic pathways or require extensive oracle calls to perform goal-directed molecular optimization. Furthermore, these *de novo* generation methods directly generate molecules in the synthesizable space exhibit poor explorability and optimization performance against the target chemical properties. An alternative approach is projecting given molecules into the synthesizable space to suggest synthesizable analogs (Gao et al., 2021; Luo et al., 2024; Gao et al., 2025; Sun et al., 2025). However, these methods are unable to reliably suggest synthetic pathways that reconstruct synthesizable molecules. All of these results indicate a lack of explorability in the synthesizable chemical space, largely due to their insufficient reasoning capability on the generated synthetic pathways.

**Retrosynthesis planning.** A related area is retrosynthesis planning, which aims to predict synthetic pathways of given synthesizable molecules in a top-down direction (Coley et al., 2019; Genheden et al., 2020; Kim et al., 2024; Sathyanarayana et al., 2025). Yu et al. (2024) addressed another problem, double-ended starting material-constrained synthesis planning, with bidirectional node expansion in AND-OR graph search. Retrosynthesis planning methods, however, cannot suggest synthesizable analogs given unsynthesizable molecules, because the target molecule is an invariable starting point for the search. In addition, since they are typically a combination of a single-step reaction prediction model and a search algorithm, they cannot consider the entire pathway to optimize the end products for target chemical property. In contrast, ReaSyn solves the problem of synthesizable projection that can be considered as *loose* single-ended synthesis planning. The target molecule guides synthesis rather than being a strict constraint, allowing flexible handling of the target molecule regardless of its synthesizability and enabling optimization of the entire pathway based on the end molecule's properties.

## 3 BACKGROUND

**Problem definition.** A set of building blocks $\mathcal{B}$ and a set of reactions $\mathcal{R}$ together define a *synthesizable chemical space*. A reaction $R \in \mathcal{R}$ is a function that maps reactants to a product, and the synthesizable space is defined as the set of product molecules that can be formed by the iterative application of reactions on compatible combinations of building blocks in $\mathcal{B}$. A reaction rule is described by SMILES Arbitrary Target Specification (SMARTS) (Daylight Chemical Information Systems, 2019), a regular expression-like representation for Simplified Molecular Input Line Entry System (SMILES) (Weininger, 1988) patterns of reactants and products. A reaction step is defined by converting the matching pattern in the reactant SMILES to a specified pattern in the product SMILES (e.g., using RDKit (Landrum et al., 2016)), and the synthetic pathway $\boldsymbol{p}$ is defined as a specific ordering of these reaction steps. Given the target molecule $\boldsymbol{x}$ as input, the goal of synthesizable projection is to generate the pathway $\boldsymbol{p}$ that produces the end product molecule that is similar to $\boldsymbol{x}$:

$$\boldsymbol{p}^* = \underset{\boldsymbol{p}}{\operatorname{argmax}} \operatorname{sim}(\operatorname{prod}(\boldsymbol{p}), \boldsymbol{x}), \tag{1}$$

where sim is the molecular similarity and prod is the function that gets the end product by executing $\boldsymbol{p}$.

**Edit Flow (EF) for sequence generation.** Recently, Havasi et al. (2025) proposed a new discrete flow model to overcome the fixed-length problem of discrete diffusion models while preserving their advantage of processing full sequences. Edit Flow defines a Continuous-Time Markov Chain (CTMC) (Campbell et al., 2024) at the full sequence level rather than at the token level. The CTMC transports sequences from a source (e.g., noise) distribution $p(\boldsymbol{p})$ to a target (e.g., data) distribution $q(\boldsymbol{p})$ via edit operations: token insertions, deletions, and substitutions. Its CTMC transition is defined as edit rates, $u_t^\theta(\operatorname{ins}(\boldsymbol{p}, i, a)|\boldsymbol{p})$, $u_t^\theta(\operatorname{del}(\boldsymbol{p}, i)|\boldsymbol{p})$, and $u_t^\theta(\operatorname{sub}(\boldsymbol{p}, i, a)|\boldsymbol{p})$ for the insertion, deletion, and substitution operations, respectively, where $\theta$ denotes the model parameters. Here, $i$ is the token index where the operations are performed, and $a$ is the token to be inserted or substituted.

The distribution of source ($\boldsymbol{p}_0$) and target ($\boldsymbol{p}_1$) sequence pairs is called *coupling* $\pi(\boldsymbol{p}_0, \boldsymbol{p}_1)$, whose marginals are $p$ and $q$, i.e., $\sum_{\boldsymbol{p}_0} \pi(\boldsymbol{p}_0, \boldsymbol{p}_1) = q(\boldsymbol{p}_1)$ and $\sum_{\boldsymbol{p}_1} \pi(\boldsymbol{p}_0, \boldsymbol{p}_1) = p(\boldsymbol{p}_0)$. Edit Flow uses the empty coupling where $\boldsymbol{p}_0$ is an empty sequence or the uniform coupling where $p(\boldsymbol{p}_0)$ is uniform over

tokens. Another important component in Edit Flow training is *alignment*, which is the process of aligning $p_0$ and $p_1$ to identify a set of edit operations that transform $p_0$ to $p_1$. Given the aligned $(p_0, p_1)$ pairs, Edit Flow can be trained using a Bregman divergence loss. More details are in Section B.

## 4 METHOD

The traverse direction is the key decision in generation of tree-structured synthetic pathways (Figure 3(a)). Most existing synthetic pathway generation methods adopt bottom-up (BU) generation, which starts with building blocks and progressively predicts more complex molecules toward the target product molecule. Although the BU approach guarantees starting from valid building blocks in $\mathcal{B}$, it inevitably faces a harder problem than top-down (TD) generation because (1) it needs to search in $\mathcal{B}$ rather than $\mathcal{R}$ during the most challenging initial few steps, where $|\mathcal{B}| \gg |\mathcal{R}|$ (e.g., 211,220 vs. 115), and (2) it is easier to reason backward from the given target molecule. However, although TD solves an easier task and is in line with how chemists infer pathways, it lacks the guarantee of reaching legitimate building blocks included in $\mathcal{B}$ at leaf nodes, making it a less popular choice in existing synthetic pathway generation methods. The distinct characteristics of the two approaches highlight the need for a new bidirectional framework that can complement and integrate them.

To this end, we introduce ReaSyn, a synthesizable projection framework that integrates BU decoding, TD decoding, and holistic editing. We start with introducing the bidirectional pathway representation of ReaSyn in Section 4.1. Next, we describe the bidirectional iterative cycle of ReaSyn in Section 4.2. Finally, we describe the holistic refinement scheme of ReaSyn using Edit Bridge in Section 4.3.

### 4.1 REPRESENTING SYNTHETIC PATHWAYS IN BOTTOM-UP AND TOP-DOWN DIRECTIONS

We first introduce a new sequential representation that can represent synthetic pathways in both BU and TD directions. Prior works (Luo et al., 2024; Gao et al., 2025) represent a synthetic pathway $p$ using a post-order traversal of its corresponding synthetic tree (Figure 7(a)) and autoregressively generate $p$ instead of directly generating the SMILES of the product molecule (Segler et al., 2018a). Unlike in the previous notation, synthetic trees are traversed in both post-order and *reverse* post-order in ReaSyn's notation. While post-order traversal yields the BU pathway representation (i.e., from leaves to root), the reverse post-order yields the TD pathway representation (i.e., from root to leaves) by reversing the post-order-traversed sequence (Figure 3(a) and Figure 7(b)). Specifically, a BU pathway sequence $p_{\text{BU}}$ consisting of $B$ blocks (i.e., subsequences), where each block $b \in \{1, \ldots, B\}$ represents either a molecule or a reaction, is represented as $p_{\text{BU}} := p^1 \oplus p^2 \oplus \cdots \oplus p^B$. Here, $\oplus$ denotes the concatenation operation and $p^b$ denotes the $b$-th block of $p$. The TD sequence of the same pathway $p_{\text{TD}}$ is then represented as $p_{\text{TD}} := p^B \oplus p^{B-1} \oplus \cdots \oplus p^1$. Molecular blocks are represented by SMILES with delimiter tokens indicating the start and end of the block ([MOL:START] and [MOL:END]), while reaction blocks consist of a single token indicating the reaction type. These *complementary* sequences, $p_{\text{BU}}$ and $p_{\text{TD}}$, use a single unified token vocabulary for molecular and reaction blocks. More details on the pathway representation and examples of $p_{\text{BU}}$ and $p_{\text{TD}}$ are provided in Section C.1.

Not only can the new notation represent pathways in both directions, but it also resolves drawbacks of pathway representations in existing methods. First, existing representations use molecular fingerprints to represent building blocks to handle large $\mathcal{B}$, but since the mapping between a molecule and its fingerprints is not bijective, there exists information loss in processing building blocks. In addition, molecular fingerprints are sparse and a mistake in a single entry in the fingerprints can result in a significantly different molecule. Our notation eliminates these problems by directly representing building blocks using SMILES, which is a smoother representation than molecular fingerprints. Secondly, existing methods employ hierarchical representations that first determine the node/token type (i.e., building block or reaction) and then predict specific node/token features (i.e., building block fingerprints or the reaction class). This scheme is prone to error accumulation and requires additional architectural complexity. For example, Luo et al. (2024) uses separate classifier heads for token type, reaction, and building block fingerprints, and Gao et al. (2025) uses a Bernoulli diffusion head for building block fingerprints. Our notation circumvents these problems by using a unified vocabulary.

## 4.2 Reasoning Synthetic Pathways in Bottom-up and Top-down Directions

**Bidirectional training and inference.** Based on the developed notations $p_{\text{BU}}$ and $p_{\text{TD}}$, we propose a simple yet effective training and inference scheme that does not require additional computational costs. Specifically, we adopt an encoder-decoder Transformer (Vaswani et al., 2017) where the encoder encodes $x$ and the decoder aims to autoregressively generate $p$. The dataset $\mathcal{D}$ consists of $(x, p)$ pairs, and we randomly switch between $p = p_{\text{BU}}$ and $p = p_{\text{TD}}$ with a probability of $p = 0.5$ during training to equip a single autoregressive model with the ability to handle both directions. The model is trained with the next token prediction loss. Importantly, since molecular blocks consist of multiple SMILES tokens while reaction blocks (and [START]/[END]) consist of a single token, we introduce a loss weighting scheme based on token type:

$$\mathcal{L} = - \mathop{\mathbb{E}}_{\substack{(x,p)\sim\mathcal{D} \\ p\sim\{p_{\text{BU}},p_{\text{TD}}\}}} \left[ \frac{1}{|\mathcal{I}_{\text{mol}}|} \sum_{i\in\mathcal{I}_{\text{mol}}} \log \pi_\theta(p_i|x, p_{1:i-1}) + \frac{1}{|\mathcal{I}_{\text{other}}|} \sum_{j\in\mathcal{I}_{\text{other}}} \log \pi_\theta(p_j|x, p_{1:j-1}) \right], \quad (2)$$

where $\mathcal{I}_{\text{mol}}$ and $\mathcal{I}_{\text{other}}$ are the sets of token indices of molecular blocks and token indices of other blocks, respectively, and $p_i$ denotes the $i$-th token of $p$. This balances the learning of synthesis pathways in accordance with the number of building blocks and reactions in them. During inference, we introduce a simple scheme to control the direction. We first note that $p_{\text{BU}}$ starts with the [MOL:START] token whereas $p_{\text{TD}}$ starts with a reaction token. To enforce a particular BU or TD direction, we simply bias the categorical distribution for the first token to sample from [MOL:START] for BU and a reaction for TD sequences. The bias is simply enforced by setting the log probabilities (e.g., logits) of non-reaction tokens to $-\infty$. This simple bidirectional training/inference scheme allows ReaSyn to perform bidirectional sampling with a single model while achieving performance comparable to two standard unidirectional models (Table 6 and Table 7). Training and inference details are included in Section D and E.

**Reasoning with bidirectional iterative cycles.** Based on the bidirectional model capable of generating pathways in both directions, we propose to complement and integrate BU and TD approaches through an iterative cycle that alternates between the two sampling directions (Figure 2). The iterative approach is designed to walk through the vast synthesizable space to discover analogs. Given the target product molecule, the cycle starts with generating an initial synthetic pathway $p_{\text{BU}}$ of $B$ blocks in a BU direction. Next, a block index $b$ is randomly drawn from $\{1, \dots, B-1\}$, and starting from the $b$-th block, ReaSyn predicts the right-hand subsequence of its complementary sequence $p_{\text{TD}}^{>b}$ $(= p_{\text{BU}}^{\leq(B-b)})$ again in a TD direction. This allows the generated synthetic tree to be refined at the subtree level. This cycle can be run multiple times until a pathway is found for a molecule sufficiently similar to the input target molecule. Searching in the synthetic tree space is challenging because nodes in the tree are interconnected with each other, and the bidirectional iterative cycle ensures that updates made at any node in the tree propagate properly all the way to the root and leaf nodes.

## 4.3 Refining Synthetic Pathways in Holistic View with Edit Bridge

The bidirectional iterative cycle that combines BU and TD generation enables deepened reasoning on synthetic pathways. ReaSyn takes this test-time search strategy one step further with Edit Flow (Havasi et al., 2025). Edit Flow suggests edits at the entire sequence level to transport the sequence toward the target distribution via edit operations (Eq. (4)). Taking full advantage of its holistic and flexible nature, we propose adding a step to the cycle that provides another perspective on the generation.

We propose Edit Bridge, an extension of Edit Flow that forms a bridge from a pretrained distribution towards the data distribution. Recall that Edit Flows build couplings from the source sequence $p_0$ to the target sequence $p_1$. In the original Edit Flows, $p_0$ is assumed to be an empty or uniformly random sequence which has no or minimally random overlap with $p_1$. Instead, in this paper, we form a coupling between a sample generated by our bidirectional autoregressive model (Section 4.2) and the target sequence $p_1$. Specifically, we assume that $p_0$ is generated via the aforementioned generative cycle of the autoregressive model and train Edit Flow to generate the target $p_1$ using coupling formed between the two. We term this model Edit Bridge as it bridges between the autoregressive model and data distribution. As shown in Table 8, our Edit Bridge coupling shows much higher $p_0$-$p_1$ alignment rate than the empty or uniform couplings (70.6% vs. 0.0% or 2.6%). Consequently, it requires much less edit operations to convert $p_0$ to $p_1$ than the two couplings, resulting in much less sampling steps during inference

Table 1: **Synthesizable molecule reconstruction results.** The results are the means and the standard deviations of 3 runs. The best results are highlighted in bold.

| Dataset | Method | Reconstruction rate (%) | Similarity | Div. (Pathway) | Div. (BB) |
|---|---|---|---|---|---|
| Enamine | SynNet (Gao et al., 2021) | $25.2 \pm 0.1$ | $0.661 \pm 0.000$ | $0.014 \pm 0.001$ | $0.239 \pm 0.002$ |
| | SynFormer (Gao et al., 2025) | $66.3 \pm 0.6$ | $0.913 \pm 0.001$ | $0.101 \pm 0.001$ | $0.587 \pm 0.002$ |
| | ReaSyn (ours) | $\mathbf{95.0} \pm 0.0$ | $\mathbf{0.987} \pm 0.001$ | $\mathbf{0.118} \pm 0.002$ | $\mathbf{0.753} \pm 0.004$ |
| ChEMBL | SynNet (Gao et al., 2021) | $7.9 \pm 0.0$ | $0.542 \pm 0.000$ | $0.009 \pm 0.000$ | $0.090 \pm 0.001$ |
| | SynFormer (Gao et al., 2025) | $19.7 \pm 0.4$ | $0.668 \pm 0.002$ | $0.039 \pm 0.000$ | $0.192 \pm 0.002$ |
| | ReaSyn (ours) | $\mathbf{31.7} \pm 0.3$ | $\mathbf{0.751} \pm 0.001$ | $\mathbf{0.050} \pm 0.000$ | $\mathbf{0.321} \pm 0.002$ |
| ZINC1k | SynNet (Gao et al., 2021) | $12.6 \pm 0.1$ | $0.456 \pm 0.001$ | $0.001 \pm 0.000$ | $0.089 \pm 0.000$ |
| | SynFormer (Gao et al., 2025) | $18.0 \pm 1.2$ | $0.624 \pm 0.003$ | $0.020 \pm 0.000$ | $0.181 \pm 0.001$ |
| | ReaSyn (ours) | $\mathbf{87.9} \pm 0.2$ | $\mathbf{0.958} \pm 0.003$ | $\mathbf{0.071} \pm 0.001$ | $\mathbf{0.658} \pm 0.002$ |

Table 2: **Synthesizable molecule reconstruction results** with the ChEMBL test set in Luo et al. (2024). The results of SynNet (Gao et al., 2021) and ChemProjector (Luo et al., 2024) are taken from Luo et al. (2024).

| Method | Reconstruction rate (%) | Sim. (Morgan) | Sim. (Scaffold) | Sim. (Gobbi) |
|---|---|---|---|---|
| SynNet (Gao et al., 2021) | 5.4 | 0.427 | 0.417 | 0.268 |
| ChemProjector (Luo et al., 2024) | 13.4 | 0.616 | 0.603 | 0.564 |
| SynFormer (Gao et al., 2025) | $19.5 \pm 0.2$ | $0.698 \pm 0.002$ | $0.673 \pm 0.002$ | $0.643 \pm 0.003$ |
| ReaSyn (ours) | $\mathbf{33.0} \pm 0.2$ | $\mathbf{0.762} \pm 0.002$ | $\mathbf{0.754} \pm 0.001$ | $\mathbf{0.722} \pm 0.001$ |

(30.0 vs. 94.6 or 142.9). In our paper, Edit Bridge is implemented using the same encoder-decoder Transformer (Vaswani et al., 2017) architecture as the autoregressive model, with additional heads to predict the edit operation rates $u_t^\theta(\text{ins}(\boldsymbol{p}, i, a)|\boldsymbol{p})$, $u_t^\theta(\text{del}(\boldsymbol{p}, i)|\boldsymbol{p})$, and $u_t^\theta(\text{sub}(\boldsymbol{p}, i, a)|\boldsymbol{p})$ (Eq. (4)).

Since both (1) the sampling process of $\boldsymbol{p}_0$ and (2) the alignment process that computes the edit operations converting $\boldsymbol{p}_0$ to $\boldsymbol{p}_1$ are computationally expensive, we prepare 10.5M training data (i.e., $(\boldsymbol{p}_0, \boldsymbol{p}_1)$ pairs and their align operations) offline. Further details are included in Section D. Unlike autoregressive generation, Edit Bridge allows ReaSyn to holistically consider the entire synthetic pathway. As shown in Figure 8 and Figure 9, Edit Bridge plays a significant role in refining pathways by jointly editing the tree skeleton and semantics. We summarize ReaSyn's single iterative refinement cycle consisting of (1) bottom-up decoding, (2) top-down decoding, and (3) holistic editing in Algorithm 1.

## 5 EXPERIMENTS

Following Gao et al. (2025), we adopt the set of 115 reactions that include common uni-, bi- and tri-molecular reactions, and the set of 211,220 purchasable building blocks in the Enamine's U.S. stock catalog retrieved in October 2023 (Enamine, 2023), together covering a synthesizable chemical space broader than $10^{60}$ molecules. The details on the model architecture are included in Section C.2.

### 5.1 SYNTHESIZABLE MOLECULE RECONSTRUCTION

**Setup.** Even given a vast synthesizable space, previous synthesizable molecule generation methods have struggled to cover this space extensively. We evaluate the coverage of ReaSyn in the synthesizable chemical space with synthesizable molecule reconstruction, where the goal is to reconstruct given molecules by proposing synthetic pathways. Following Gao et al. (2025), the reconstruction is evaluated on randomly selected 1,000 molecules from the Enamine REAL diversity set (Enamine, 2023) and ChEMBL (Gaulton et al., 2012). In addition, to simulate common real-world scenarios in which the set of purchasable building blocks expands after the model is trained, we include another challenging benchmark by expanding the building block set with unseen building blocks. Concretely, we add 37,386 molecules with fewer than 18 heavy atoms from ZINC250k (Irwin et al., 2012), yielding a total of 248,606 building blocks. 1,000 molecules generated using the defined reaction rules and the new building blocks are used as the test set, which we denote as ZINC1k. We also experiment with another test set of ChEMBL molecules introduced by Luo et al. (2024). Further details are included in Section E.1.

**Metrics.** Following Gao et al. (2025), **reconstruction rate**, the fraction of generated pathways that yield the identical product molecule to the input molecule, and **similarity**, the average Tanimoto similarity on the Morgan fingerprint (Morgan, 1965; Rogers & Hahn, 2010) between the generated molecule and the input molecule, are used to evaluate the model. We further report **diversity (pathway)**, the average molecular diversity of end products from analogous pathways, and **diversity**

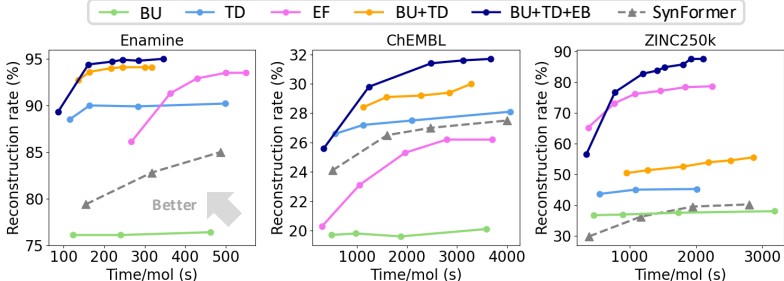

Figure 4: **Ablation study on synthesizable molecule reconstruction.** SynFormer uses the beam width of 96, exhaustiveness of 256, and multiple cycles.

**(BB)**, the average diversity of unique building blocks in analogous pathways, to assess diversity of generation. Here, *analogous pathways* are defined as pathways that yield a product molecule with a Tanimoto similarity $\geq 0.8$ to the input molecule. For the experiment with the test set of Luo et al. (2024), **similarity** between the input target molecule and the projected analog is measured using the Tanimoto similarity on three molecular fingerprints: (1) Morgan fingerprint (Morgan, 1965), (2) Morgan fingerprint of Murcko scaffold, and (3) Gobbi pharmacophore fingerprint (Gobbi & Poppinger, 1998), following Luo et al. (2024).

**Results.** In Table 1 and Table 2, we compare ReaSyn with state-of-the-art synthesizable analog generation methods, SynNet (Gao et al., 2021) and SynFormer (Gao et al., 2025). ReaSyn significantly outperforms the baselines across all metrics. Specifically, ReaSyn shows a high reconstruction rate and similarity, demonstrating that it successfully reconstructs input molecules by extensively exploring a synthesizable chemical space. It also shows high diversity in Table 1, demonstrating its ability to propose diverse pathways. Note that unlike the Enamine and ZINC1k test sets, molecules in the ChEMBL test set may lie beyond the defined synthesizable space (i.e., may require reactions or building blocks beyond $\mathcal{B}$ and $\mathcal{R}$), so the performance is generally low on ChEMBL. Notably, ReaSyn outperforms the prior works by a particularly large margin on the ZINC1k test set, which simulates challenging scenarios with out-of-distribution test molecules and unseen building blocks. This result highlights the strong generalizability of ReaSyn in generating out-of-distribution synthetic pathways.

## 5.2 ABLATION STUDY

To examine the effect of each component in ReaSyn's generative cycle, i.e., (1) **BU** decoding, (2) **TD** decoding, and (3) Edit Bridge (**EB**) editing, we conduct ablation studies on the synthesizable molecule reconstruction task (Section 5.1) in Figure 4. The cycle is proposed as an effective method for scaling test-time compute, and there is generally a trade-off between reconstruction rate and sampling time. BU, TD, and EF are ReaSyns that each use only one component of the cycle. Note that since the EB coupling requires a pretrained source distribution, EF instead uses the empty coupling. BU+TD is ReaSyn that uses the bidirectional iteration in the cycle, and BU+TD+EB is the complete ReaSyn that leverages all the three components. First, comparing BU+TD to BU or TD, we observe that using the autoregressive model in a bidirectional way shows much better performance than unidirectional sampling. The unidirectional schemes fail to reconstruct a large portion of the test molecules even when the test-time compute is increased, demonstrating the importance of the proposed bidirectional iteration. Secondly, we can reconfirm the effectiveness of the bidirectional iteration when we compare BU+TD+EB to EF. Lastly, we observe significant improvements with the additional refinement step using EB, when we compare BU+TD+EB to BU+TD. Overall, these results together highlight the importance of leveraging multiple perspectives in synthetic pathway generation.

## 5.3 SYNTHESIZABLE GOAL-DIRECTED MOLECULAR OPTIMIZATION

While goal-directed molecular optimization methods aim to address essential drug discovery tasks, their practicality is limited as the generated molecules are often not synthesizable (Gao & Coley, 2020). To overcome this problem, synthesizable projection can be applied to goal-directed molecular optimization. Similar to Gao et al. (2025), we demonstrate the application of ReaSyn in exploring the local synthesizable space in conjunction with an off-the-shelf molecular optimization method. Specifically, we use synthesizable projection of ReaSyn as an additional mutation operator of a genetic algorithm (GA) by projecting the offspring molecules after every reproduction step of Graph

Table 3: **Synthesizable goal-directed molecular optimization results on the TDC oracles.** The results are the means of AUC top-10 and average top-10 SA scores of 3 runs. The results for the baselines other than Graph GA-SF and Graph GA are taken from Sun et al. (2025). The best synthesis-based results are highlighted in bold.

| Method | Graph GA-ReaSyn | Graph GA-SF | SynthesisNet | SynNet | DoG-Gen | DoG-AE | Graph GA |
|---|---|---|---|---|---|---|---|
| Synthesis | ✓ | ✓ | ✓ | ✓ | ✓ | ✓ | ✗ |
| amlodipine_mpo | 0.620 | **0.696** | 0.608 | 0.567 | 0.537 | 0.509 | 0.651 |
| celecoxib_rediscovery | **0.810** | 0.559 | 0.582 | 0.443 | 0.466 | 0.357 | 0.682 |
| drd2 | **0.977** | 0.972 | 0.960 | 0.969 | 0.949 | 0.944 | 0.970 |
| fexofenadine_mpo | 0.788 | 0.786 | **0.791** | 0.764 | 0.697 | 0.681 | 0.785 |
| gsk3b | **0.889** | 0.803 | 0.848 | 0.790 | 0.832 | 0.602 | 0.838 |
| jnk3 | **0.695** | 0.658 | 0.639 | 0.631 | 0.596 | 0.470 | 0.693 |
| median1 | 0.274 | **0.308** | 0.305 | 0.219 | 0.218 | 0.172 | 0.261 |
| median2 | **0.259** | 0.258 | 0.257 | 0.237 | 0.213 | 0.183 | 0.257 |
| osimertinib_mpo | **0.823** | 0.816 | 0.810 | 0.797 | 0.776 | 0.751 | 0.829 |
| perindopril_mpo | **0.561** | 0.530 | 0.524 | 0.559 | 0.475 | 0.433 | 0.533 |
| ranolazine_mpo | **0.752** | 0.751 | 0.741 | 0.743 | 0.712 | 0.690 | 0.745 |
| sitagliptin_mpo | 0.314 | **0.338** | 0.313 | 0.026 | 0.048 | 0.010 | 0.524 |
| zaleplon_mpo | 0.460 | 0.478 | **0.528** | 0.341 | 0.123 | 0.050 | 0.458 |
| Average score | **0.633** | 0.612 | 0.608 | 0.545 | 0.511 | 0.450 | **0.633** |

GA (Jensen, 2019), thus ensuring all the resulting offspring molecules lie in the synthesizable space. We denote the resulting GA as Graph GA-ReaSyn. We emphasize that this approach is universal and other molecular optimization methods can also be used. Further details are included in Section E.2.

### 5.3.1 OPTIMIZATION OF TDC ORACLES

**Setup.** Following Sun et al. (2025), we conduct 15 goal-directed molecular optimization tasks of the benchmark of Gao et al. (2022) that simulate real-world drug discovery with the TDC oracle functions (Brown et al., 2019; Huang et al., 2021), and use the **AUC top-10** to assess the optimization performance.

**Results.** The results are shown in Table 3. We compare Graph GA-ReaSyn with synthesis-based baselines that restrict the generation to the synthesizable chemical space. We also include Graph GA (Jensen, 2019) to examine the impact of the synthesizable projection. Graph GA-ReaSyn outperforms all synthesis-based baselines in optimization performance, showing its effectiveness in discovering chemical optima in the synthesizable space. Notably, it achieves comparable optimization performance to Graph GA even with the synthesizability constraint, validating that the proposed synthesizable projection can generate synthesizable analogs while maintaining the core molecular properties.

### 5.3.2 OPTIMIZATION OF sEH BINDING AFFINITY

**Setup.** Following Cretu et al. (2025), we conduct optimization of the binding affinity against the protein target soluble epoxide hydrolase (sEH), measured by a pretrained proxy model (Bengio et al., 2021). As the evaluation metrics, the average **sEH** binding affinity, synthetic accessibility (**SA**) score (Ertl & Schuffenhauer, 2009), quantitative estimate of drug-likeness (**QED**) (Bickerton et al., 2012), and the **AiZynthFinder** success rate (Genheden et al., 2020) of generated molecules are reported. AiZynthFinder is a widely used retrosynthesis planning method, and its success rate is considered a more reliable synthesizability proxy than SA score.

**Results.** The results are shown in Table 4. FragGFN (Bengio et al., 2021) and SynFlowNet (Cretu et al., 2025) are GFlowNets (Bengio et al., 2021) with a fragment action space and a reaction action space, respectively, and SyntheMol (Swanson et al., 2024) is a property predictor-guided MCTS method. '(SA)' or '(QED)' denote using a modified optimization objective with SA score or QED. All baselines except FragGFN take the approach of constraining the design space to the synthesizable space. As shown in the table, FragGFN shows a very poor SA score because it does not consider synthesizability. ReaSyn outperforms previous methods in terms of all the metrics, verifying that synthesizable projection using ReaSyn is a more effective strategy to find synthesizable chemical optima.

### 5.4 SYNTHESIZABLE HIT EXPANSION

**Setup.** Synthesizable projection of ReaSyn can suggest multiple synthesizable analogs for a given target molecule with a beam search, and thus can be applied to hit expansion to find synthesizable

Table 4: **Synthesizable goal-directed molecular optimization results on the sEH proxy.** The results are the means and the standard deviations of 3 runs. The results for the baselines are taken from Cretu et al. (2025). The best results are highlighted in bold.

| Method | Syn. | sEH | SA $\downarrow$ | QED | AiZynth. |
|---|---|---|---|---|---|
| FragGFN | ✗ | $0.77 \pm 0.01$ | $6.28 \pm 0.02$ | $0.30 \pm 0.01$ | $0.00$ |
| FragGFN (SA) | ✗ | $0.70 \pm 0.01$ | $5.45 \pm 0.05$ | $0.29 \pm 0.01$ | $0.00$ |
| SyntheMol | ✓ | $0.64 \pm 0.01$ | $3.08 \pm 0.01$ | $0.63 \pm 0.01$ | $0.82$ |
| SynFlowNet | ✓ | $0.92 \pm 0.01$ | $2.92 \pm 0.01$ | $0.59 \pm 0.02$ | $0.65$ |
| SynFlowNet (SA) | ✓ | $0.94 \pm 0.01$ | $2.67 \pm 0.03$ | $0.68 \pm 0.01$ | $0.93$ |
| SynFlowNet (QED) | ✓ | $0.86 \pm 0.03$ | $4.02 \pm 0.26$ | $0.74 \pm 0.04$ | $0.55$ |
| Graph GA-ReaSyn | ✓ | $\mathbf{0.96} \pm 0.00$ | $\mathbf{2.05} \pm 0.01$ | $\mathbf{0.75} \pm 0.01$ | $\mathbf{0.97} \pm 0.01$ |

Table 5: **JNK3 hit expansion results.** The results are the means and the standard deviations of 3 runs. The best results are highlighted in bold.

| Method | Analog rate (%) | Improve rate (%) | Success rate (%) |
|---|---|---|---|
| SynNet | $13.7 \pm 0.1$ | $1.2 \pm 0.0$ | $1.0 \pm 0.0$ |
| SynFormer | $31.4 \pm 1.9$ | $6.0 \pm 0.3$ | $4.1 \pm 0.3$ |
| ReaSyn (ours) | $\mathbf{75.7} \pm 1.8$ | $\mathbf{11.8} \pm 0.4$ | $\mathbf{8.8} \pm 0.7$ |

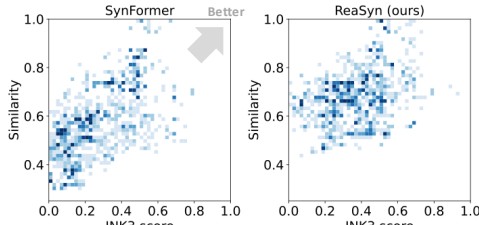

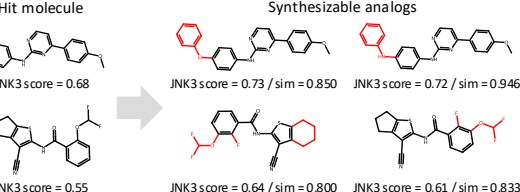

Figure 5: **The distribution of JNK3 scores and analog similarity** of SynFormer and ReaSyn.

Figure 6: **Examples of generated synthesizable analogs from JNK3 hit expansion.** Modified substructures in the analogs are indicated by red.

analogs of hit molecules. Following Gao et al. (2025), we conduct hit expansion on the identified hit molecules for c-Jun NH2-terminal kinase 3 (JNK3) inhibition. Specifically, the proxy from the TDC library (Huang et al., 2021) is used as the oracle function that scores the inhibition of JNK3, and the top-10 scoring molecule from ZINC250k (Irwin et al., 2012) are selected as the hits. The JNK3 proxy is set as the reward model to guide the search and 100 analogs are generated for each hit, yielding a total of 1,000 synthesizable molecules. **Analog rate**, the fraction of generated unique analogs that have Tanimoto similarities $\geq 0.6$ to the input hit, **improve rate**, the fraction of generated unique analogs that have higher JNK3 scores than the original hit, and **success rate**, the fraction of generated unique analogs that satisfy both, are computed by averaging over the generated molecules.

**Results.** The results are shown in Table 5. ReaSyn exhibits superior performance to the previous synthesizable projection methods in terms of all the metrics. The high analog rate shows that ReaSyn can broadly search the synthesizable space to suggest close analogs, while the high improve rate shows that ReaSyn can find chemical optima in terms of the target property thanks to the goal-directed search. We also provide the distribution of suggested molecules in Figure 5 and examples in Figure 6. Compared to SynFormer, ReaSyn generates synthesizable molecules with high similarities and high JNK3 inhibition scores, highlighting its effectiveness in hit expansion.

## 6 CONCLUSION

A major weakness of molecular generative models is the generation of synthetically inaccessible molecules. In our paper, we introduced ReaSyn, an effective framework for synthesizable projection with deepened reasoning on synthetic pathways. The iterative refinement cycle of (1) bottom-up decoding, (2) top-down decoding, and (3) holistic editing constitutes a powerful pathway reasoning strategy from multiple viewpoints. ReaSyn showed superior performance on a variety of synthesis-constrained drug discovery tasks. These results demonstrate ReaSyn's strong applicability as a tool to navigate combinatorially-large synthesizable chemical space in real-world drug discovery. Considering higher-level compatibility in ReaSyn, such as selectivity or functional groups, is left as future work. To facilitate research in this space, we will release code and models publicly.

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

# Appendix

## A    ITERATIVE REFINEMENT CYCLE OF REASYN

---
**Algorithm 1** A Single Generation Cycle of ReaSyn
---

**Input:** Trained bidirectional autoregressive model $M_{\text{BUTD}}$, trained Edit Bridge model $M_{\text{EB}}$,
       the input target molecule $x$, the number of pathways to refine with Edit Bridge $N_{\text{EB}}$
Set $\mathcal{P} \leftarrow \{\}$
▷ *Bottom-up generation (refinement)*
Set $p_{\text{BU,start}} \leftarrow [[\texttt{START}], [\texttt{MOL:START}]]$
**with beam search**
     Generate $p_{\text{BU}}$ by sampling and appending tokens to $p_{\text{BU,start}}$ using $M_{\text{BUTD}}$ given $x$
     Set $\mathcal{P} \leftarrow \mathcal{P} \cup \{p_{\text{BU}}\}$
**end beam search**
Calculate scores of $p \in \mathcal{P}$ as $\text{sim}(\text{prod}(p), x)$ (Eq. (6))
▷ *Top-down refinement*
**with beam search**
     Sample $p$ from $\mathcal{P}$ based on their scores
     Get $p_{\text{TD}}$, the TD representation of $p$
     Randomly draw a block index $b$ from the block indices of $p_{\text{TD}}$
     Repredict $p_{\text{TD}}^{>b}$ by sampling using $M_{\text{BUTD}}$ given $x$
       (**if** $p_{\text{TD}}^{>b} = [[\texttt{START}]]$ **then** set the non-reaction logits to $-\infty$)
     Set $\mathcal{P} \leftarrow \mathcal{P} \cup \{p_{\text{TD}}\}$
**end beam search**
Calculate scores of $p \in \mathcal{P}$ as $\text{sim}(\text{prod}(p), x)$ (Eq. (6))
▷ *Edit Bridge refinement*
**for** $i = 1, \ldots, N_{\text{EB}}$ **do**
     Sample $p$ from $\mathcal{P}$ based on their scores
     Generate $p_{\text{EB}}$ by refining $p$ using $M_{\text{EB}}$
     Set $\mathcal{P} \leftarrow \mathcal{P} \cup \{p_{\text{EB}}\}$
**end for**
Calculate scores of $p \in \mathcal{P}$ as $\text{sim}(\text{prod}(p), x)$ (Eq. (6))
**Output:** Generated pathways $\mathcal{P}$ and their similarity scores with $x$

---

We summarize a single refinement cycle of ReaSyn in Algorithm 1.

## B    OVERVIEW OF EDIT FLOW

Recently, Edit Flow (Havasi et al., 2025), has been proposed to overcome the fixed-variable length problem of discrete diffusion models in sequence generation. By defining a discrete flow on sequences, Edit Flow enjoys flexible and position-relative sequence generation. Specifically, Edit Flow defines a Continuous-Time Markov Chain (CTMC) (Campbell et al., 2024) at the full sequence level rather than at the token level.

A CTMC transports sequences from a source (e.g., noise) distribution $p(p)$ to a target (e.g., data) distribution $q(p)$, and this transition is characterized by a rate $u_t$. The distribution of source ($p_0$) and target ($p_1$) sequence pairs is called the coupling $\pi(p_0, p_1)$ distribution, whose marginals are $p$ and $q$[2]:

$$\sum_{p_0} \pi(p_0, p_1) = q(p_1), \qquad \sum_{p_1} \pi(p_0, p_1) = p(p_0). \tag{3}$$

Edit Flow in the original paper uses the empty coupling where $p_0$ is an empty sequence or the uniform coupling where $p(p_0)$ is uniform over tokens. These couplings are independent couplings,

---
[2]A sequence is denoted as $x$ in the original paper, but we instead use $p$ to maintain consistent notation with the synthetic pathway sequences.

i.e., $\pi(\boldsymbol{p}_0, \boldsymbol{p}_1) = p(\boldsymbol{p}_0)q(\boldsymbol{p}_1)$, and the source sequence $\boldsymbol{p}_0$ does not contain information about the target sequence $\boldsymbol{p}_1$.

Edit Flow models the transition via edit operations: token insertions, deletions, and substitutions. The CTMC rate $u_t^\theta$, where $\theta$ denotes the model parameters, is defined as follows:

$$
\begin{aligned}
u_t^\theta(\mathrm{ins}(\boldsymbol{p}, i, a)|\boldsymbol{p}) &= \lambda_{t,i}^{\mathrm{ins}}(\boldsymbol{p})Q_{t,i}^{\mathrm{ins}}(a|\boldsymbol{p}) && \text{for } i \in \{1, \ldots, n(\boldsymbol{p})\} \\
u_t^\theta(\mathrm{del}(\boldsymbol{p}, i)|\boldsymbol{p}) &= \lambda_{t,i}^{\mathrm{del}}(\boldsymbol{p}) && \text{for } i \in \{1, \ldots, n(\boldsymbol{p})\} \\
u_t^\theta(\mathrm{sub}(\boldsymbol{p}, i, a)|\boldsymbol{p}) &= \lambda_{t,i}^{\mathrm{sub}}(\boldsymbol{p})Q_{t,i}^{\mathrm{sub}}(a|\boldsymbol{p}) && \text{for } i \in \{1, \ldots, n(\boldsymbol{p})\}
\end{aligned}
\tag{4}
$$

where $\boldsymbol{p}$ is a sequence of length $n(\boldsymbol{p})$ and $\mathrm{ins}(\boldsymbol{p}, i, a)$, $\mathrm{del}(\boldsymbol{p}, i)$, and $\mathrm{sub}(\boldsymbol{p}, i, a)$ denote the insertion, substitution, and deletion operations, respectively. $i$ is the token index where the operations are performed, and $a$ is the token to be inserted or substituted. $\lambda_{t,i} \geq 0$ are the total rates of inserting, deleting, or substituting any token at $i$. $Q_{t,i}$ are the distributions over tokens given insertion or substitution occurs at position $i$.

There exist multiple possible sets of edit operations that transition from $\boldsymbol{p}_0$ to $\boldsymbol{p}_1$. To handle this, Edit Flow introduces an auxiliary sequence $\boldsymbol{z}$ that additionally has a special blank token $\varepsilon$. The process of identifying the set of edit operations using $\boldsymbol{z}$ is called the alignment process. Given the pair $(\boldsymbol{p}_0, \boldsymbol{p}_1)$, the aligned pair is denoted as $(\boldsymbol{z}_0, \boldsymbol{z}_1)$, where $\boldsymbol{z}_0$ and $\boldsymbol{z}_1$ have the same length. As an example, if $(\boldsymbol{p}_0, \boldsymbol{p}_1)$ is ('kitten', 'smitten'), the optimal $(\boldsymbol{z}_0, \boldsymbol{z}_1)$ is ('k$\varepsilon$itten', 'smitten'). Edit operations can be easily recovered given the aligned pair as the token conversion $a \to b$ is an insertion if $a = \varepsilon$, a deletion if $b = \varepsilon$, or a substitution if $a \neq \varepsilon$ and $b \neq \varepsilon$.

Given the aligned pairs, Edit Flow can be trained using a Bregman divergence loss:

$$
\mathcal{L}(\theta) = \mathbb{E}_{\substack{\pi(\boldsymbol{z}_0, \boldsymbol{z}_1) \\ t, p_t(\boldsymbol{p}_t, \boldsymbol{z}_t | \boldsymbol{z}_0, \boldsymbol{z}_1)}} \left[ \sum_{\boldsymbol{p} \neq \boldsymbol{p}_t} u_t^\theta(\boldsymbol{p}|\boldsymbol{p}_t) - \sum_{i=1}^{N} \mathbb{1}_{[\boldsymbol{z}_1^i \neq \boldsymbol{z}_t^i]} \frac{\dot{\kappa}_t}{1 - \kappa_t} \log u_t^\theta(\boldsymbol{p}(\boldsymbol{z}_t, i, \boldsymbol{z}_1^i)|\boldsymbol{p}_t) \right], \tag{5}
$$

where $\boldsymbol{p}(\boldsymbol{z}_t, i, \boldsymbol{z}_1^i)$ is the operation that substitutes the $i$-th token in $\boldsymbol{z}_t$ with $\boldsymbol{z}_1^i$ and then removes all $\varepsilon$ in $\boldsymbol{z}_t$, which corresponds to one edit operation of Eq. (4).

For a more detailed explanation on Edit Flow, please refer to the original paper (Havasi et al., 2025).

## C  DETAILS ON PATHWAY REPRESENTATION AND MODEL

### C.1  BIDIRECTIONAL PATHWAY NOTATION

Prior works (Luo et al., 2024; Gao et al., 2025) represent a synthetic pathway $\boldsymbol{p}$ using a post-order traversal of its corresponding synthetic tree (Figure 7(a)). This representation captures synthesis as a bi-level sequence consists of the token type level and the token feature level. At the token type level, each token is one of four types: [START], [BB] (building block), [RXN] (reaction), or [END]. At the token feature level, different embedding strategies are used based on the token types: [BB] tokens are embedded using Morgan fingerprints since the set of purchasable building blocks is large and depends on purchasable stock catalogs, while [RXN] and other fixed tokens use the standard lookup embeddings. To match this bi-level representation, inference is performed hierarchically: a classifier first predicts the token type, then depending on the outcome, either a fingerprint network predicts the building block or a classifier selects the reaction. This representation can describe both linear and convergent synthetic pathways, and has been shown effective in bottom-up synthesis planning (Luo et al., 2024; Gao et al., 2025), especially due to its compatibility with autoregressive generation.

However, this notation and other previous notations on synthetic pathways (1) use molecular fingerprints to represent building blocks to handle large $\mathcal{B}$ and (2) are hierarchical (Bradshaw et al., 2020; Gao et al., 2021; Swanson et al., 2024; Luo et al., 2024; Gao et al., 2025; Koziarski et al., 2024; Cretu et al., 2025; Seo et al., 2025; Sun et al., 2025). These characteristics have several drawbacks, therefore we have proposed a new representation that directly uses SMILES, is non-hierarchical, and is able to represent pathways in both bottom-up and top-down directions (Section 4.1).

Our bidirectional notation is parsed using a unified token vocabulary of size 272, including [START], [END], [MOL:START], [MOL:END], 153 SMILES tokens and 115 reaction tokens. The SMILES tokens include the following:

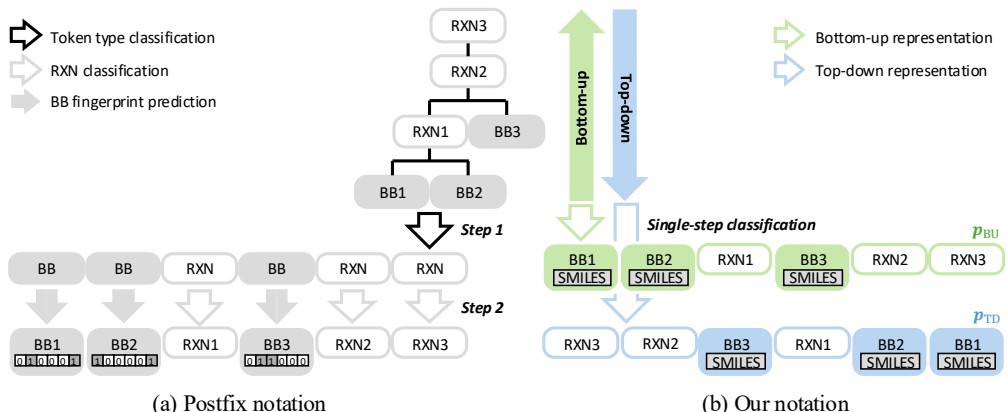

(a) Postfix notation                   (b) Our notation

Figure 7: **Comparison of the postfix notation and our notation.** `[START]` and `[END]` tokens are omitted for simplicity. The postfix notation (Luo et al., 2024; Gao et al., 2025) represents a synthetic pathway with a bi-level sequence consists of the token type level and the token feature level (i.e., the Morgan fingerprints for `[BB]` tokens and the reaction class for `[RXN]` tokens). Consequently, the synthetic pathways are embedded and generated in a hierarchical way in each autoregressive step. In contrast, our proposed notation uses a unified vocabulary without hierarchy to represent the complementary sequences $p_{\mathrm{BU}}$ and $p_{\mathrm{TD}}$.

```
H, He, Li, Be, B, C, N, O, F, Ne, Na, Mg, Al, Si, P, S, Cl, Ar,
K, Ca, Sc, Ti, V, Cr, Mn, Fe, Co, Ni, Cu, Zn, Ga, Ge, As, Se, Br,
Kr, Rb, Sr, Y, Zr, Nb, Mo, Tc, Ru, Rh, Pd, Ag, Cd, In, Sn, Sb, Te,
I, Xe, Cs, Ba, La, Ce, Pr, Nd, Pm, Sm, Eu, Gd, Tb, Dy, Ho, Er, Tm,
Yb, Lu, Hf, Ta, W, Re, Os, Ir, Pt, Au, Hg, Tl, Pb, Bi, Po, At, Rn,
Fr, Ra, Ac, Th, Pa, U, Np, Pu, Am, Cm, Bk, Cf, Es, Fm, Md, No, Lr,
Rf, Db, Sg, Bh, Hs, Mt, Ds, Rg, Cn, Nh, Fl, Mc, Lv, Ts, Og, b, c,
n, o, s, p, 0, 1, 2, 3, 4, 5, 6, 7, 8, 9, [, ], (, ), ., =, #, -,
+, \, /, :, ~, @, ?, >, *, $, %
```

The input target molecules are also encoded using the above tokens.

We provide examples of $p$ in the bidirectional notation. An example synthetic pathway of `[N-]=[N+]=NCC1C[N+](C(=O)C(CC(=O)O)C2CCCO2)=C(N)N1c1cccc(F)c1` (the target molecule of Figure 8) is as follows:

$p_{\mathrm{BU}}$ = `[START]` `[MOL:START]` `[N-]=[N+]=NCC1C[NH+]=C(N)N1c1cccc(F)c1` `[MOL:END]` `[MOL:START]` `O=C1CC(C2CCCO2)C(=O)O1` `[MOL:END]` `[RXN:44]` `[END]`

$p_{\mathrm{TD}}$ = `[START]` `[RXN:44]` `[MOL:START]` `[N-]=[N+]=NCC1C[NH+]=C(N)N1c1cccc(F)c1` `[MOL:END]` `[MOL:START]` `O=C1CC(C2CCCO2)C(=O)O1` `[MOL:END]` `[END]`

An example synthetic pathway of `Cc1ccc(S(=O)(=O)c2ccc(C(CO)CCOCC(F)(F)F)cc2)c(OCC2CO2)c1` (the target molecule of Figure 9) is as follows:

$p_{\mathrm{BU}}$ = `[START]` `[MOL:START]` `OCC(CCOCC(F)(F)F)c1ccc(Cl)cc1` `[MOL:END]` `[MOL:START]` `Cc1ccc(Cl)c(OCC2CO2)c1` `[MOL:END]` `[RXN:32]` `[END]`

$p_{\mathrm{TD}}$ = `[START]` `[RXN:32]` `[MOL:START]` `OCC(CCOCC(F)(F)F)c1ccc(Cl)cc1` `[MOL:END]` `[MOL:START]` `Cc1ccc(Cl)c(OCC2CO2)c1` `[MOL:END]` `[END]`

Here, special tokens other than SMILES tokens and reaction tokens are displayed in gray. Note that the first token after the `[START]` token (the first token sampled during inference) is always `[MOL:START]` and a reaction token in $p_{\mathrm{BU}}$ and $p_{\mathrm{TD}}$, respectively.

## C.2 Model Architecture

ReaSyn has two models, one for the bidirectional autoregressive pathway generation and one for the Edit Bridge pathway editing. For both models, we adopt the standard encoder-decoder Transformer architecture, following Gao et al. (2025). Specifically, the encoder has 6 layers, with a hidden dimension of 768, 16 attention heads, a feed-forward dimension of 4096, and the maximum sequence length of 256. The decoder has 10 layers, with a hidden dimension of 768, 16 attention heads, a feed-forward dimension of 4096, and the maximum sequence length of 512. Overall, the autoregressive model has 166M parameters and the Edit Bridge model has 174M parameters. The Edit Bridge model has three additional heads for $\lambda_{t,i} := (\lambda_{t,i}^{\text{ins}}, \lambda_{t,i}^{\text{del}}, \lambda_{t,i}^{\text{sub}})$, $Q_{t,i}^{\text{ins}}$, and $Q_{t,i}^{\text{sub}}$, respectively (Eq. (4)). Note that ReaSyn's models have fewer parameters than SynFormer (Gao et al., 2025) of 230M parameters, because they do not require heads for fingerprint diffusion and reaction classification.

## D Details on Training

In our paper, following Gao et al. (2025), we used the set of 211,220 purchasable building blocks in the Enamine's U.S. stock catalog retrieved in October 2023 (Enamine, 2023), and the set of 115 reactions that include common uni-, bi- and tri-molecular reactions curated by Gao et al. (2025)[3]. Given the sets of building blocks and reactions $\mathcal{B}$ and $\mathcal{R}$, the dataset of $\boldsymbol{p}$ can be generated on-the-fly during training by iteratively executing compatible reactions to random building blocks. A stack is maintained for each pathway to keep track of intermediate products during data generation, and the corresponding token blocks of building blocks and reactions are concatenated to construct a sequence $\boldsymbol{p}$ as described in Section 4.1.

ReaSyn's autoregressive model was trained using the dataset of $(\boldsymbol{x}, \boldsymbol{p})$ pairs with the next token prediction loss (Eq. (2)). The AdamW optimizer (Loshchilov & Hutter, 2019) with a learning rate of $3e-4$ for 500k steps was used. A batch size per GPU was set to 64, and 8 NVIDIA A100 GPUs were used. The training took about 5 days.

We implement Edit Bridge based on the codebase[4]. ReaSyn's Edit Bridge model was trained using the dataset of $(\boldsymbol{x}, \boldsymbol{z}_0, \boldsymbol{z}_1)$ triplets. The dataset was generated offline using our proposed Edit Bridge coupling. Specifically, given a $(\boldsymbol{x}, \boldsymbol{p}_1)$ pair where $\boldsymbol{p}_1$ corresponds to the true pathway of molecule $\boldsymbol{x}$, we first generate $\boldsymbol{p}_0$ with the autoregressive model pretrained with the training procedure described above. In this paper, we used $\boldsymbol{p}_{\text{BU}}$ without the bidirectional iterative cycle as $\boldsymbol{p}_0$. Then the aligned $(\boldsymbol{z}_0, \boldsymbol{z}_1)$ pair is obtained via the alignment process (explained in Section B). Generating 10.5M data points with 120 NVIDIA A100 GPUs took about 3 days. Using the dataset, the Edit Bridge model was trained with the Bregman loss objective (Eq. (5)). The AdamW optimizer (Loshchilov & Hutter, 2019) with a learning rate of $3e-4$ for 500k steps was used. A batch size per GPU was set to 128, and 8 NVIDIA A100 GPUs were used.

## E Details on Inference

The proposed bidirectional iterative cycle explained in Section 4.2 and Algorithm 1 can be viewed as a variant of Gibbs sampling (Geman & Geman, 1984). To sample a synthetic pathway $\boldsymbol{p}$ from the distribution $p(\boldsymbol{p}) = p(\boldsymbol{p}^1, \ldots, \boldsymbol{p}^B)$, ReaSyn repeats the process of uniformly drawing the block index $b \sim \{1, \ldots, B-1\}$ and (1) sampling $\boldsymbol{p}^{>b}$ from the distribution $p(\boldsymbol{p}^{>b}|\boldsymbol{p}^{\leq b})$ or (2) sampling $\boldsymbol{p}^{\leq b}$ from the distribution $p(\boldsymbol{p}^{\leq b}|\boldsymbol{p}^{>b})$. This is equivalent to performing Gibbs sampling with the kernels $p(\boldsymbol{p}^{>b}|\boldsymbol{p}^{\leq b})$ and $p(\boldsymbol{p}^{\leq b}|\boldsymbol{p}^{>b})$, allowing ReaSyn to extensively search the neighborhood of the synthetic tree space.

Inference of ReaSyn was implemented similarly to the official codebase[5] of Gao et al. (2025). Specifically, the Transformer decoder of ReaSyn autoregressively generates a pathway block-by-block, and a stack is maintained for each pathway $\boldsymbol{p}$ being generated. If the generated block $\boldsymbol{p}^b$ corresponds to a molecule ($\boldsymbol{x}'$), a building block is retrieved from the building block set $\mathcal{B}$ by conducting the nearest-neighbor search, and then pushed to the stack. Here, the nearest-neighbor

---

[3] https://github.com/wenhao-gao/synformer (Apache-2.0 license)
[4] https://github.com/TheMatrixMaster/edit-flows-demo
[5] https://github.com/wenhao-gao/synformer (Apache-2.0 license)

search uses the following similarity function:

$$\text{sim}(\boldsymbol{x}, \boldsymbol{x}') = \frac{1}{\text{dist}(\boldsymbol{x}, \boldsymbol{x}') + 0.1}, \tag{6}$$

where $\text{dist}(\boldsymbol{x}, \boldsymbol{x}')$ is the 1-norm distance between the Morgan fingerprints (Morgan, 1965) (radius 2, length 2048) of $\boldsymbol{x}$ and $\boldsymbol{x}'$ if $\boldsymbol{x}'$ is valid. If $\boldsymbol{x}'$ is an invalid SMILES, $\text{dist}(\boldsymbol{x}, \boldsymbol{x}')$ is calculated as the character-wise edit distance between the SMILES strings of $\boldsymbol{x}$ and $\boldsymbol{x}'$. If the generated block $\boldsymbol{p}^b$ is a reaction, the required number of reactants are popped from the top of the stack, then the reaction is executed using the RDKit (Landrum et al., 2016) library. Unlike in Gao et al. (2025), if the predicted reation is incompatible with the reactants in the current stack, ReaSyn chooses the next reaction with the next highest prediction logit. The resulting intermediate product is pushed back to the stack.

To extensively explore the synthesizable chemical space with multiple synthesizable analog suggestions, beam search is employed. Following previous works (Luo et al., 2024; Gao et al., 2025), ReaSyn employs beam search which tracks the top-scoring stacks being generated and expands them block-by-block. Molecular blocks are scored by the aforementioned similarity used in the building block retrieval step (Eq. (6)). Reaction blocks are scored by the classification probability predicted by the model, and a stack's score is defined as the sum of the scores of its blocks.

All experiments were conducted using 4 NVIDIA A100 GPUs.

### E.1 SYNTHESIZABLE MOLECULE RECONSTRUCTION

Similarity in Table 1 and similarity (Morgan) in Table 2 are measured as the Tanimoto similarity on the Morgan fingerprint (Morgan, 1965) with radius 2 and length 4096. We used Therapeutics Data Commons (TDC) library (Huang et al., 2021) to calculate diversity. In Figure 4, for BU, TD, BU+TD, and BU+TD+EB, the test-time compute was scaled using different values of number of the bidirectional iterative cycles. The search width, exhaustiveness, and number of Edit Bridge samples in each cycle were set to 8, 4, and 100, respectively. The best values of BU+TD+EB were also reported in Table 1, which used 12, 24, and 16 cycles for Enamine, ChEMBL, and ZINC250k, respectively. We used the default search width of 64 and exhaustiveness of 24 for ChemProjector (Luo et al., 2024) and SynFormer (Gao et al., 2025), and used a search width of 12 for SynNet (Gao et al., 2021).

### E.2 SYNTHESIZABLE GOAL-DIRECTED MOLECULAR OPTIMIZATION

In Table 3, a population size of 100 and an offspring size of 100 were used. The rate of the mutation operation of Graph GA was set to 0.1. Following Gao et al. (2022) and Sun et al. (2025), the maximum number of oracle calls was set to 10k the optimization performance was evaluated with the area under the curve (AUC) of the top-10 average score versus oracle calls. We performed the grid search (search width) $\in \{1, 2\}$ for each oracle function. For drd2, gsk3b, perindopril_mpo, sitagliptin_mpo, and zaleplon_mpo, the search width was set to 1; for all others, it was set to 2. The exhaustiveness, the number of bidirectional cycle, and the number of Edit Bridge samples for cycle was set to 4, 1, and 4 for all oracle functions.

In Table 4, a population size of 400 and an offspring size of 100 were used. The rate of the mutation operation of Graph GA was set to 0.1. The search width, exhaustiveness, the number of bidirectional cycle, and the number of Edit Bridge samples for cycle was set to 2, 4, 1, and 4, respectively. The objective function to optimize is set to sEH score $\cdot \widehat{\text{SA}} \cdot \widehat{\text{QED}}$, where sEH score is the normalized binding affinity of the sEH protein target predicted by a proxy model (Bengio et al., 2021), and $\widehat{\text{SA}}$ and $\widehat{\text{QED}}$ are normalized SA score (Ertl & Schuffenhauer, 2009) and QED (Bickerton et al., 2012) defined as:

$$\widehat{\text{SA}} = \frac{10 - \text{SA}}{9} \in [0, 1], \quad \widehat{\text{QED}} = 2 \cdot \text{clip}(\text{QED}, 0, 0.5) \in [0, 1]. \tag{7}$$

We used Therapeutics Data Commons (TDC) library (Huang et al., 2021) to calculate SA and QED. Note that in Table 4, the baselines used a total of 300k oracle calls and the average values of 1k generated molecules were reported. In contrast, Graph GA-ReaSyn used only 5k oracle calls and the average values of top-1k generated molecules generated among the 5k molecules were reported, showing very high sampling efficiency compared to the baselines.

Table 6: **Reconstruction rate (%) results in synthesizable molecule reconstruction** with different train/inference schemes.

| Train | Inference | Dataset | | |
|---|---|---|---|---|
| | | Enamine | ChEMBL | ZINC250k |
| BU | BU | 75.3 | 22.5 | 51.3 |
| TD | TD | 85.2 | 25.6 | 41.4 |
| BU+TD | BU | 78.3 | 22.4 | 38.8 |
| BU+TD | TD | 82.5 | 25.4 | 44.0 |

Table 7: **Synthesizable molecule reconstruction results** of ReaSyn that uses two separate autoregressive models for BU and TD and ReaSyn that uses a single autoregressive model that does both BU and TD sampling.

| Dataset | Method | Reconstruction rate (%) | Similarity | Div. (Pathway) | Div. (BB) |
|---|---|---|---|---|---|
| Enamine | ReaSyn-separate BU+TD | 95.2 | 0.987 | 0.118 | 0.757 |
| | ReaSyn | 95.0 | 0.987 | 0.118 | 0.753 |
| ChEMBL | ReaSyn-separate BU+TD | 31.4 | 0.750 | 0.050 | 0.321 |
| | ReaSyn | 31.7 | 0.751 | 0.050 | 0.321 |
| ZINC250k | ReaSyn-separate BU+TD | 88.1 | 0.960 | 0.074 | 0.681 |
| | ReaSyn | 87.9 | 0.958 | 0.071 | 0.658 |

Table 8: **Comparison of different couplings of Edit Flow**. The values are the average of 10,000 random training data.

| Coupling | Aligned rate (%) ↑ | # of edit ops./seq. ↓ | Rate of align operations (%) | | |
|---|---|---|---|---|---|
| | | | Insertion | Deletion | Substitution |
| Empty (Havasi et al., 2025) | 0.00 | 94.55 | 100.00 | 0.00 | 0.00 |
| Uniform (Havasi et al., 2025) | 2.59 | 142.87 | 14.75 | 36.90 | 48.35 |
| Edit Bridge (ours) | **70.56** | **30.01** | 26.75 | 25.24 | 48.01 |

### E.3 SYNTHESIZABLE HIT EXPANSION

In the synthesizable hit expansion experiment, The search width, exhaustiveness, the number of bidirectional cycle, and the number of Edit Bridge samples for cycle was set to 12, 128, 12, and 100, respectively. Up to 100 synthesizable analogs were collected for each input molecule.

## F ADDITIONAL EXPERIMENTAL RESULTS

### F.1 COMPARISON OF UNIDIRECTIONAL AND BIDIRECTIONAL TRAINING

We compared the standard unidirectional and our proposed bidirectional training/inference scheme of ReaSyn's autoregressive model in Table 6. The top two rows (BU train, TD train) are the models that use a single fixed direction during training and inference. The bottom two rows (BU+TD train) correspond to the model that uses the bidirectional training/inference scheme explained in Section 4.2. Note that the BU train and TD train are separate models, while BU+TD train is a single model that can do both BU and TD inference. As shown in the table, the bidirectional model shows no significant difference in performance compared to the unidirectional models despite using only a single checkpoint.

This result can be reconfirmed in Table 7. ReaSyn-separate BU+TD indicates the method that uses two autoregressive models (BU train and TD train in Table 6) to implement the bidirectional iterative cycle, and ReaSyn indicates the method that only uses a single autoregressive model (BU+TD train in Table 6). As shown in the table, the proposed bidirectional learning/inference scheme exhibits no significant performance difference compared to the unidirectional method, with much higher memory efficiency.

### F.2 COMPARISON OF DIFFERENT COUPLINGS IN EDIT FLOW

We provide a comparison of different couplings in Edit Flow in Table 8. Empty coupling and uniform coupling are the proposed in the original paper (Havasi et al., 2025). Since these are independent

Table 9: **Reconstruction rate (%) results in synthesizable molecule reconstruction** of AiZynthFinder (Genheden et al., 2020) and ReaSyn. The results are the means and the standard deviations of 3 runs. The best results are highlighted in bold.

| Method | Dataset | | |
|---|---|---|---|
| | Enamine | ChEMBL | ZINC250k |
| AiZynthFinder (Genheden et al., 2020) | $34.0 \pm 0.3$ | **$55.1 \pm 0.1$** | $11.4 \pm 0.1$ |
| AiZynthFinder (our BBs) | $79.2 \pm 0.4$ | $44.3 \pm 0.2$ | $20.5 \pm 0.2$ |
| ReaSyn (ours) | **$95.0 \pm 0.0$** | $31.7 \pm 0.3$ | **$87.9 \pm 0.2$** |

couplings, $p_1$ does not contain information about $p_0$, resulting in low aligned rates and requiring many edit operations. On the contrary, the proposed Edit Bridge coupling starts from $p_0$ which is already partially aligned with $p_1$ (aligned rate fo 70.56%), therefore requires significantly fewer edit operations (30.01).

### F.3 COMPARISON WITH RETROSYNTHESIS PLANNING METHOD

Assuming all input molecules are synthesizable, the synthesizable analog generation problem can be applied to the problem of retrosynthesis planning. Since the pathways that successfully reconstruct a given input molecule in synthesizable analog generation correspond to the solved pathways in retrosynthesis planning, we compare the reconstruction rates of ReaSyn with those of a state-of-the-art retrosynthesis planning method in Table 9. AiZynthFinder (Genheden et al., 2020) is an MCTS-based method that recursively breaks down a given product molecule into reactant molecules to find pathways to building blocks. Note that it uses 42,554 reaction templates extracted from the USPTO reaction set (Lowe, 2017) and 17,422,831 building blocks from the ZINC stock (Irwin et al., 2012), so its solution space is much larger than that of ReaSyn, which is defined by 115 reactions and 211,220 building blocks in this paper. We also include AiZynthFinder results which uses the same building blocks as ReaSyn. We used the official codebase[6] to run AiZynthFinder.

As shown in the table, ReaSyn shows higher reconstruction rate on the Enamine and ZINC250k test sets and lower reconstruction rate on the ChEMBL test set. We suspect that this is because the synthesizable space of AiZynthFinder is more in line and compatible with the ChEMBL test set than that of ReaSyn. Nevertheless, ReaSyn outperforms AiZynthFinder by a large margin in the other two test sets, despite its much smaller design space. Notably, ReaSyn outperforms AiZynthFinder by a particularly large margin on the ZINC250k test set, which requires out-of-distribution generalization on unseen building blocks. Retrosynthesis planning methods adopt a TD approach which sequentially infers simpler molecules to arrive at building blocks, making them unsuitable for inferring pathways that consider out-of-distribution building blocks. Moreover, we emphasize that the synthesizable analog generation approach is much more versatile. Only reconstruction rate can be measured using retrosynthesis planning methods, and they cannot be applied to other tasks, such as suggesting synthesizable analogs given unsynthesizable molecules, goal-directed optimization of the end products, or synthesizable hit expansion.

### F.4 GENERATED EXAMPLES

We provide examples of reconstructed Enamine molecules in synthesizable molecule reconstruction in Figure 8 and Figure 9. In generating these examples, the search width, exhaustiveness, number of bidirectional cycles, and number of Edit Bridge samples in each cycle were set to 2, 4, 1, and 100, respectively, and ZINC250k building blocks were included. We also provide additional examples of hit molecules and generated synthesizable analogs in JNK3 hit expansion (Section 5.4) in Figure 10.

---

[6]https://github.com/MolecularAI/aizynthfinder (MIT license)

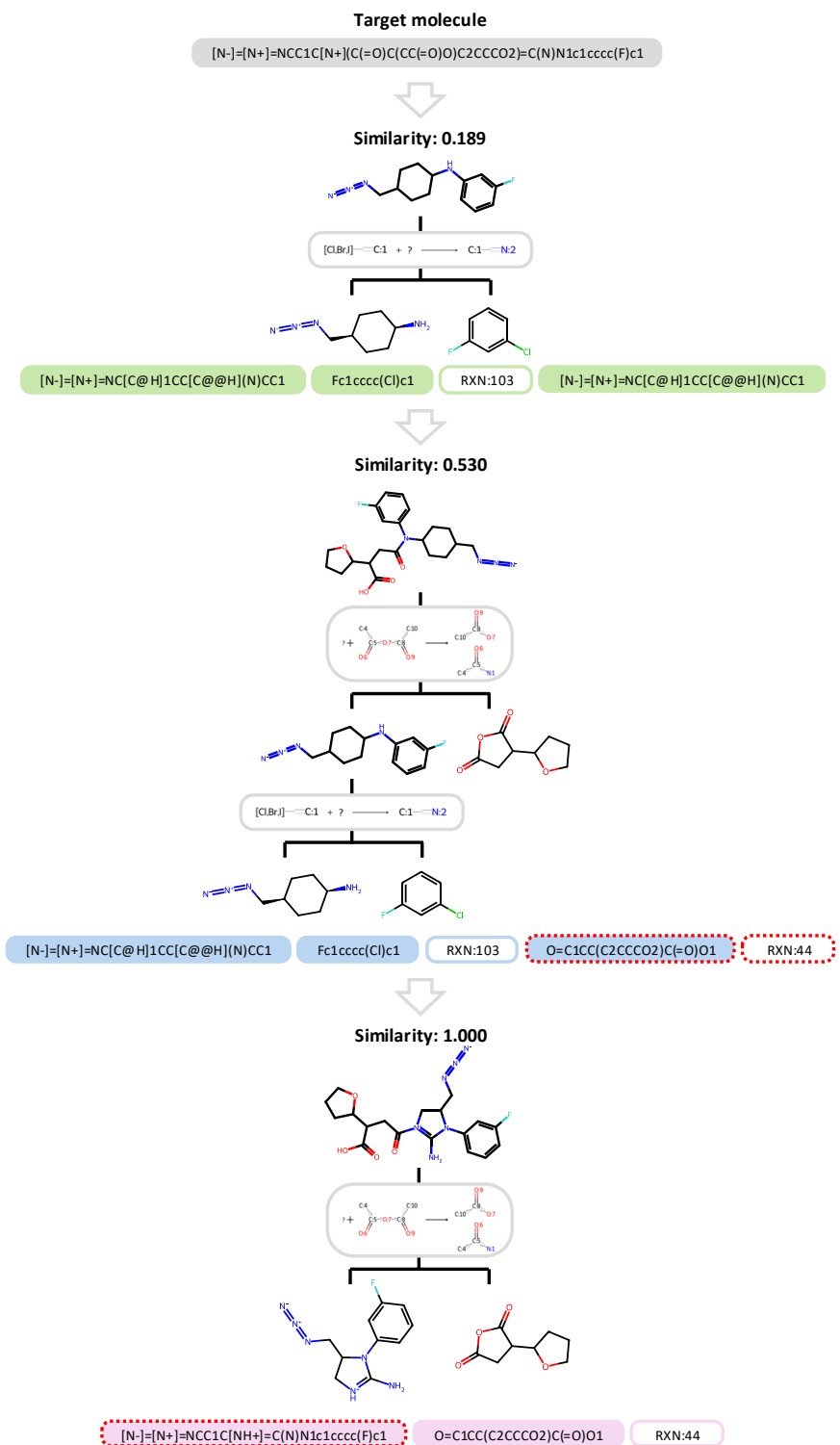

Figure 8: **Examples of ReaSyn's generation cycle.** All sequences are represented in a bottom-up order.

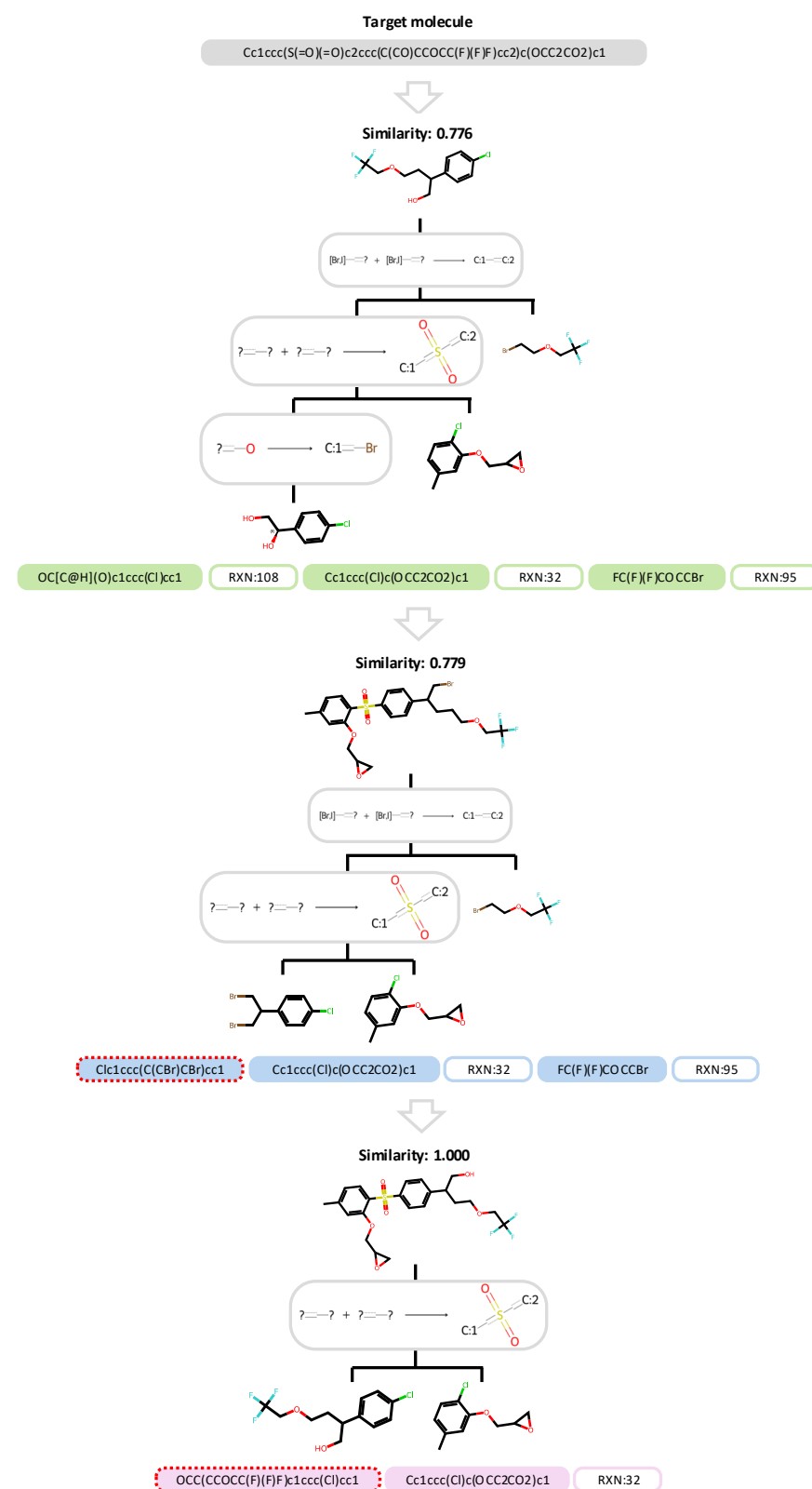

Figure 9: **Examples of ReaSyn's generation cycle (continued).** All sequences are represented in a bottom-up order.

| Hit molecule | Synthesizable analogs | | | | |
|---|---|---|---|---|---|
| JNK3 score = 0.68 | JNK3 score = 0.74
sim = 0.674 | JNK3 score = 0.73
sim = 0.850 | JNK3 score = 0.73
sim = 0.684 | JNK3 score = 0.73
sim = 0.633 | JNK3 score = 0.72
sim = 0.773 |
| JNK3 score = 0.67 | JNK3 score = 0.67
sim = 0.971 | JNK3 score = 0.64
sim = 0.872 | JNK3 score = 0.64
sim = 0.850 | JNK3 score = 0.56
sim = 0.810 | JNK3 score = 0.56
sim = 0.810 |
| JNK3 score = 0.66 | JNK3 score = 0.63
sim = 0.923 | JNK3 score = 0.57
sim = 0.787 | JNK3 score = 0.56
sim = 0.686 | JNK3 score = 0.50
sim = 0.667 | JNK3 score = 0.48
sim = 0.667 |
| JNK3 score = 0.62 | JNK3 score = 0.57
sim = 0.804 | JNK3 score = 0.51
sim = 0.482 | JNK3 score = 0.51
sim = 0.482 | JNK3 score = 0.49
sim = 0.673 | JNK3 score = 0.48
sim = 0.673 |
| JNK3 score = 0.55 | JNK3 score = 0.64
sim = 0.800 | JNK3 score = 0.63
sim = 0.786 | JNK3 score = 0.63
sim = 0.786 | JNK3 score = 0.61
sim = 0.833 | JNK3 score = 0.60
sim = 0.959 |
| JNK3 score = 0.51 | JNK3 score = 0.61
sim = 0.745 | JNK3 score = 0.55
sim = 0.604 | JNK3 score = 0.53
sim = 0.756 | JNK3 score = 0.53
sim = 0.756 | JNK3 score = 0.52
sim = 0.667 |
| JNK3 score = 0.50 | JNK3 score = 0.53
sim = 0.875 | JNK3 score = 0.53
sim = 0.875 | JNK3 score = 0.53
sim = 0.875 | JNK3 score = 0.52
sim = 0.961 | JNK3 score = 0.52
sim = 0.961 |
| JNK3 score = 0.50 | JNK3 score = 0.79
sim = 0.704 | JNK3 score = 0.79
sim = 0.691 | JNK3 score = 0.79
sim = 0.691 | JNK3 score = 0.78
sim = 0.704 | JNK3 score = 0.69
sim = 0.576 |
| JNK3 score = 0.49 | JNK3 score = 0.92
sim = 0.696 | JNK3 score = 0.88
sim = 0.727 | JNK3 score = 0.63
sim = 0.714 | JNK3 score = 0.59
sim = 0.683 | JNK3 score = 0.57
sim = 0.656 |
| JNK3 score = 0.49 | JNK3 score = 0.57
sim = 0.761 | JNK3 score = 0.57
sim = 0.604 | JNK3 score = 0.57
sim = 0.604 | JNK3 score = 0.57
sim = 0.739 | JNK3 score = 0.56
sim = 0.644 |

Figure 10: **Examples of hit molecules and generated synthesizable analogs** by ReaSyn in JNK3 hit expansion. JNK3 inhibition score measured by the JNK3 proxy and similarity to the input hit are provided at the bottom of each generated analog.

