# OpenReview forum: "Exploring Synthesizable Chemical Space with Iterative Pathway Refinements"
_ICLR.cc/2026/Conference — ICLR 2026 Oral_

### Official Review · Reviewer_5E2P · 2025-10-21

**Soundness:** 3
**Presentation:** 3
**Contribution:** 3
**Rating:** 8
**Confidence:** 3

**Summary:**

- This paper addresses the problem of synthesizable molecule generation, which aims to generate molecules that are synthetically accessible.
- The authors propose ReaSyn, an iterative generative pathway retirement framework based on reaction trees. It effectively generates both molecules and their corresponding synthetic pathway (i.e., how to synthesize the molecules).
- The results demonstrate superior performance compared to state-of-the-art methods for synthesizable molecule generation.

**Strengths:**

- The paper is well written and clearly structured.
- It overcomes existing difficulties in tree-structured synthetic pathway generation, which are typically tackled using bottom-up or top-down approach. By integrating bottom-up decoding, top-down decoding and Edit bridge (recently propsoed), the proposed model allows to iterative generation and refinement of the generated pathways. I think this integration is a novel contribution of this paper, even though the individual components of ReaSyn are known and commonly used in pathway generation.
- The results clearly show that ReaSyn outperforms state-of-the-art models across various evaluation metrics.

**Weaknesses:**

The generation cycle consists of 3 steps: bottom-up generation, top-down generation, and Edit bridge for pathway retirement, while the compared models use only a single step. It would be helpful to discuss how much slower ReaSyn is in terms of training and inference time (generation of pathway), compared to other methods.

**Questions:**

See the weakness.

**Details Of Ethics Concerns:**

I have no ethics concerns for this paper.

---

> ### Author Response · Authors · 2025-11-21
>
> We sincerely appreciate your comments that our paper is well-written, clearly structured, novel, and the results clearly show that our method outperforms existing SOTA methods.
> We address your question below.
>
> ---
> > The generation cycle consists of 3 steps: bottom-up generation, top-down generation, and Edit bridge for pathway retirement, while the compared models use only a single step. It would be helpful to discuss how much slower ReaSyn is in terms of training and inference time (generation of pathway), compared to other methods.
>
> The training of ReaSyn’s autoregressive/Edit Bridge model took about 5 days with 8 NVIDIA A100 GPUs as explained in lines 835-836. The training of SynFormer takes about 6~7 days with 8 NVIDIA A100 GPUs [A].
>
> We have updated Figure 4 to additionally include the comparison with SynFormer [A]. When running SynFormer, we increased the beam size to 96 and exhaustiveness to 256 (the default setting in the original paper was a beam width of 24 and exhaustiveness of 64), and ran multiple cycles to match the sampling process of ReaSyn. As shown in the figure, SynFormer falls short of ReaSyn's performance (BU+TD+EB) by a large margin. Moreover, SynFormer performs worse than ReaSyn that uses only a single direction (BU or TD), demonstrating that ReaSyn outperforms SynFormer through both (1) novel unification of bottom-up and top-down bidirectional pathway prediction and (2) component-specific contribution.
>
> ---
> *References*
>
> [A] Gao et al., Generative AI for Navigating Synthesizable Chemical Space, PNAS, 2025.
>
> ---
> We hope our response could address your question and clarify any confusion. If our response is satisfactory, we would like to kindly ask you to consider raising your score. Otherwise, we will be happy to further discuss and update the paper.

---

> > ### Comment · Reviewer_5E2P · 2025-11-27
> >
> > I thank the authors for the clarification. I would like to keep the score unchanged.

---

### Official Review · Reviewer_RDTc · 2025-10-31

**Soundness:** 3
**Presentation:** 3
**Contribution:** 3
**Rating:** 6
**Confidence:** 4

**Summary:**

The paper proposes ReaSyn, a framework for synthesizable projection that turns an arbitrary target molecule into one or more synthesizable analogs together with explicit synthetic pathways. The key ideas are:

1. a sequential pathway representation that encodes both molecules (via SMILES blocks) and reaction types in a single token vocabulary, enabling bottom-up (BU) and top-down (TD) traversals of a synthesis tree using the same autoregressive Transformer;

2. a bidirectional iterative cycle that alternates BU sampling (from building blocks up) with TD subtree re-generation (from the product down), propagating edits throughout the pathway; and

3. Edit Bridge, a discrete-flow editor over full pathway sequences (insertion/deletion/substitution) trained with offline alignments to holistically refine the pathway beyond local autoregressive edits.

On synthesizable molecule reconstruction across Enamine, ChEMBL, and a harder ZINC250k-augmented setting, ReaSyn reports markedly higher reconstruction rates, similarities, and diversities than prior synthesizable-space baselines. When plugged into Graph-GA as a projection/mutation step, ReaSyn maintains strong goal-directed optimization performance while ensuring synthesizability, and it outperforms prior synthesis-aware methods on TDC oracles, an sEH proxy (with improved SA/QED and AiZynth success), and JNK3 hit expansion (higher analog, improve, and success rates).

**Strengths:**

Unification of BU and TD in one policy. Clever direction control (first-token bias) plus balanced token-type loss yields a single model capable of both traversals; the iterative BU↔TD cycle is a natural way to propagate local edits through the pathway.

Simpler, more expressive representation. Replacing hierarchical, fingerprint-based encodings with SMILES blocks and reaction tokens reduces architectural complexity and avoids fingerprint information loss/sparsity.

Holistic refinement with Edit Bridge. A discrete-flow editor that operates at the full pathway level complements autoregressive edits and demonstrably improves reconstruction/diversity in ablations.

Strong empirical results across tasks. Large gains on reconstruction (including OOD ZINC250k building-block expansion), competitive or better optimization on TDC oracles while keeping synthesizability, improved SA/QED/AiZynth on sEH, and substantially better hit-expansion metrics for JNK3.

Thoughtful ablations. Clear attribution for (i) bidirectional iteration vs. single-direction, (ii) Edit Bridge vs. none, and (iii) representation differences vs. a close BU baseline with a near-identical Transformer.

Practical orientation. Uses a fixed reaction rule set and purchasable building-block catalog, aligning evaluation with real synthesis constraints; integrates cleanly as a projection module in standard optimization loops.

**Weaknesses:**

Compute transparency. The paper mentions a large offline alignment corpus for Edit Bridge and multi-stage decoding/beam search, but lacks wall-clock, FLOPs/tokens, and beam/budget controls, making fairness vs. baselines (and scaling laws) hard to judge.

Search-policy details. The TD subtree choice is uniform random over blocks; more informed selection might improve efficiency/quality. Ablations over the number of BU↔TD iterations vs. Edit Bridge steps at matched compute are limited.

Data hygiene & leakage. Reconstruction uses molecules drawn from the same vendor catalogs/rule families that likely seed training. Clear train/val/test separation for pathway patterns (and for the offline $(p_0,p_1)$ edit pairs) is under-specified.

Metric sensitivity. Similarity relies mainly on Morgan-Tanimoto; scaffold/pharmacophore scores are included in one benchmark, but systematic metric sensitivity and rank correlations across tasks are not fully explored.

**Questions:**

1. Catalog & rule portability. How does performance shift if (a) the building-block catalog changes (e.g., a newer Enamine snapshot or a different vendor), and (b) the reaction set is expanded/altered? Any zero-shot results with unseen rules?

2. Edit Bridge hygiene. How are $(p_0,p_1)$ pairs generated to prevent leakage from evaluation targets? Do you withhold all evaluation molecules/routes from the edit-alignment dataset?

3. Route executability. Beyond AiZynth success, can you provide route-level checks (e.g., selectivity flags, incompatible functional groups, protecting-group needs) or human/chemist audits on a subset of proposed pathways?

4. Policy control. Did you try non-uniform TD block selection (e.g., uncertainty or mismatch heuristics) and adaptive stopping for the BU↔TD loop? Any gains in sample efficiency?

5. Scaffolds vs. properties. For optimization/hit expansion, how does performance trade off between scaffold retention and property improvement as you vary the similarity threshold?

6. Ablations at fixed cost. If you fix total decoding steps, how do (i) more BU↔TD iterations vs. (ii) more Edit Bridge steps vs. (iii) larger beam affect outcomes?

7. Failure taxonomy. What are the most common failure modes (e.g., rule conflicts, unreachable leaves, unstable intermediates), and how often does Edit Bridge repair them vs. induce new ones?

---

> ### Author Response · Authors · 2025-11-21
>
> We sincerely appreciate your comments that our paper presents the unification of BU and TD through clever direction control, more effective representation, holistic refinement with Edit Bridge, strong empirical results across tasks, thoughtful ablations, and a practical approach. We address your concerns below.
>
> ---
> > Compute transparency. The paper mentions a large offline alignment corpus for Edit Bridge and multi-stage decoding/beam search, but lacks wall-clock, FLOPs/tokens, and beam/budget controls, making fairness vs. baselines (and scaling laws) hard to judge.
>
> As explained in lines 842-843, generating 10.5M data points for Edit Bridge took about 3 days with 120 NVIDIA A100 GPUs.
>
> We have reported the wall-clock sampling time/mol in Figure 4. We have updated Figure 4 to additionally include the comparison with SynFormer [A]. When running SynFormer, we increased the beam size to 96 and exhaustiveness to 256 (the default setting in the original paper was a beam width of 24 and exhaustiveness of 64), and ran multiple cycles to match the sampling process of ReaSyn. As shown in the figure, SynFormer falls short of ReaSyn's performance (BU+TD+EB) by a large margin. Moreover, SynFormer performs worse than ReaSyn, which uses only a single direction (BU or TD), demonstrating that ReaSyn outperforms SynFormer through both (1) novel unification of bottom-up and top-down bidirectional pathway prediction and (2) component-specific contribution.
>
> ---
> > The TD subtree choice is uniform random over blocks; more informed selection might improve efficiency/quality. Did you try non-uniform TD block selection (e.g., uncertainty or mismatch heuristics) and adaptive stopping for the BU↔TD loop? Any gains in sample efficiency?
>
> We sincerely appreciate your suggestion. After examining more informative TD subtree selection such as those based on molecular similarity between the intermediate product and the target molecule, we found it made little difference to the final performance, so we opted for the simpler random selection. Following SynFormer [A], we applied early stopping in sampling when ReaSyn discovered an exact pathway leading to the target molecule during the cycle.
>
> ---
> > Ablations over the number of BU↔TD iterations vs. Edit Bridge steps at matched compute are limited. If you fix total decoding steps, how do (i) more BU ↔ TD iterations vs. (ii) more Edit Bridge steps vs. (iii) larger beam affect outcomes?
>
> For the ablations over the number of BU/TD interactions vs. Edit Bridge steps, we compared them in Figure 4. As explained in lines 397-398, when we compare BU+TD+EB to BU+TD, we observed significant improvements with the additional refinement step using Edit Bridge under matched compute. The points on each line in Figure 4 correspond to results with varying numbers of cycles. We also observed that increasing the beam size has a similar effect to increasing the number of cycles. Therefore, we can conclude that adding Edit Bridge refinement is essential for further improving overall performance, rather than continuing only BU ↔ TD iterations or increasing the beam size.
>
> ---
> > Reconstruction uses molecules drawn from the same vendor catalogs/rule families that likely seed training. Clear train/val/test separation for pathway patterns is under-specified. How are $(p_0,p_1)$ pairs generated to prevent leakage from evaluation targets? Do you withhold all evaluation molecules/routes from the edit-alignment dataset?
>
> Building blocks are separated from the test set molecules; therefore, the model must predict the correct pathways that lead to the test set molecules using the building blocks. All training pathways $\boldsymbol{p}_1$ were generated on-the-fly during the training of ReaSyn's autoregressive model or the offline generation process of ReaSyn's Edit Bridge model, and thus differ from the test molecules.
>
> Moreover, we experimentally validated that ReaSyn shows excellent performance in generating synthesizable molecules as evaluated by a popular external retrosynthesis prediction tool, AiZynthFinder [B], in Table 4 and Table 9. AiZynthFinder uses 42,554 reaction templates extracted from the USPTO reaction set and 17,422,831 building blocks from the ZINC stock, so its solution space is expected to be completely different from the test molecules.

---

> ### Author Response · Authors · 2025-11-21
>
> > Similarity relies mainly on Morgan-Tanimoto; scaffold/pharmacophore scores are included in one benchmark, but systematic metric sensitivity and rank correlations across tasks are not fully explored.
>
> Following previous benchmark settings, we included a wide range of evaluation metrics across many benchmarks. These metrics include Morgan-Tanimoto similarity, scaffold/pharmacological similarity, pathway/BB diversity, target property optimization performance, SA, QED, and AiZynthFinder success rate. We emphasize that ReaSyn shows excellent performance across these metrics, validating its effectiveness as a practical tool for real-world synthesizable drug discovery problems.
>
> ---
> > How does performance shift if (a) the building-block catalog changes (e.g., a newer Enamine snapshot or a different vendor), and (b) the reaction set is expanded/altered? Any zero-shot results with unseen rules?
>
> In the synthesizable molecule reconstruction benchmark (Table 1 and Table 2), we used Enamine’s US Stock Catalog as the default building block set and expanded it with ZINC250k molecules in the ZINC250k reconstruction task. From the excellent performance of ZINC250k reconstruction results, we observe strong generalizability of ReaSyn in generating out-of-distribution synthetic pathways with unseen building blocks. Furthermore, the strong generalizability of ReaSyn is reconfirmed in the AiZynthFinder success rates shown in Table 4 and Table 9 (AiZynthFinder uses 42,554 reaction templates extracted from the USPTO reaction set and 17,422,831 building blocks from the ZINC stock, which are completely different from the reaction/building block sets used during training).
>
> ---
> > Beyond AiZynth success, can you provide route-level checks (e.g., selectivity flags, incompatible functional groups, protecting-group needs) or human/chemist audits on a subset of proposed pathways?
>
> Following previous works [A, C, D], ReaSyn does not consider higher-level compatibility such as selectivity or functional groups, leaving this for future work. We added this comment in Conclusion (lines 490-491; highlighted in green). However, ReaSyn does consider the validity of building blocks or the basic compatibility between building blocks and reaction rules, and we additionally report success rates below. Here, the **success rate** refers to the validity of the entire pathway, requiring the validity of all building block SMILES included in the pathway and building block-reaction compatibility. As shown in the table, ReaSyn shows near-perfect success rate across test sets.
>
> **Table: Synthesizable molecule reconstruction results.**
> | Dataset | Success rate (%) | Reconstruction rate (%) |
> | --- | --- | --- |
> | Enamine | 99.9 | 95.0 |
> | ChEMBL | 99.6 | 31.7 |
> | ZINC250k | 99.9 | 87.9 |
>
> ---
> > For optimization/hit expansion, how does performance trade off between scaffold retention and property improvement as you vary the similarity threshold?
>
> We used the similarity threshold of 0.6 in hit expansion following previous papers [A, C]. Here, we additionally report the hit expansion results with the similarity threshold of 0.4 as below. As shown in the table, the more generous similarity threshold leads to a much higher analog rate. With the same improve rate, this automatically translates to an improved success rate.
>
> **Table: JNK3 hit expansion results.**
> | Similarity threshold | Analog rate (%) | Improve rate (%) | Success rate (%) |
> | --- | --- | --- | --- |
> | 0.6 | 75.7 | 11.8 | 8.8 |
> | 0.4 | 99.0 | 11.8 | 11.8 |
>
> ---
> > What are the most common failure modes (e.g., rule conflicts, unreachable leaves, unstable intermediates), and how often does Edit Bridge repair them vs. induce new ones?
>
> The most common failure mode involves the generation of product molecules far from the target molecule due to incorrect reaction steps in pathways. As shown in Figures 8 and 9, ReaSyn's Edit Bridge can effectively repair such bad steps. These examples, together with Figure 4, demonstrate that holistic editing using Edit Bridge is essential for ReaSyn's superior performance.
>
> ---
> *References*
>
> [A] Gao et al., Generative AI for Navigating Synthesizable Chemical Space, PNAS, 2025.
>
> [B] Genheden et al., Aizynthfinder: a fast, robust and flexible open-source software for retrosynthetic planning. Journal of cheminformatics, 2020.
>
> [C] Luo et al., Projecting Molecules into Synthesizable Chemical Spaces, ICML, 2024.
>
> [D] Gao et al., Amortized Tree Generation for Bottom-up Synthesis Planning and Synthesizable Molecular Design, ICLR, 2021.
>
> ---
> We hope our response could address your questions and clarify any confusion. If our response is satisfactory, we would like to kindly ask you to consider raising your score. Otherwise, we will be happy to further discuss and update the paper.

---

### Official Review · Reviewer_xjoZ · 2025-11-01

**Soundness:** 3
**Presentation:** 1
**Contribution:** 2
**Rating:** 4
**Confidence:** 4

**Summary:**

This paper proposes ReaSyn, an iterative, bidirectional pathway–generation framework that combines (i) bottom-up decoding, (ii) top-down decoding, and (iii) a holistic editing step (“Edit Bridge”) to project molecules into a synthesizable space and improve coverage on reconstruction, goal-directed optimization, and hit expansion. The technical idea is sound and the empirical results are strong, but parts of the presentation make the method look more general than it actually is. If the authors clarify the core contribution, the experimental setup, and the fairness of the comparisons, I would support acceptance.

**Strengths:**

- A novel way of combining top-down, bottom-up, and holistic edit steps in one framework (effectively three networks working together).
- Effective test-time scaling: adding TD and the Edit Bridge gives non-trivial gains.
- Strong results on the harder, expanded-stock setting.

**Weaknesses:**

- Misleading Figure 1 and Table 1. The chemical space is **not** ZINC; ZINC is used to augment the building-block set. As written, the figures can easily be read as “reconstructing ZINC,” which is inaccurate. Please rename/reword.
- Time cost in Figure 4. The iterative pipeline appears pretty costly, especially for non-Enamine compounds. Please report the inference time for SynFormer in the figure/table.
- Fairness of comparison. ReaSyn’s gains partly come from using three networks and scaling at inference. Comparing this directly to single-pass baselines is somewhat unfair. The paper should clearly state the main contribution and separate the gains from architecture vs. extra inference compute.

**Questions:**

N/A

---

> ### Author Response · Authors · 2025-11-21
>
> We sincerely appreciate your comments that our paper presents a novel way of combining top-down, bottom-up, and holistic edit steps in one framework, effective test-time scaling, and strong results on the harder, expanded-stock setting. We address your concerns below.
>
> ---
> > Misleading Figure 1 and Table 1. The chemical space is not ZINC; ZINC is used to augment the building-block set. As written, the figures can easily be read as “reconstructing ZINC,” which is inaccurate. Please rename/reword.
>
> We apologize for the confusion. As explained in lines 319-320, the ZINC250k test molecules were constructed using the predefined reaction set and the ZINC250k building blocks, and we used the term ZINC250k for simplicity. We fixed the caption in Figure 1 and used a new term ‘ZINC1k’ to denote the test molecules from ZINC250k (lines 42-43, 330, 366, and 369; highlighted in green).
>
> ---
> > Time cost in Figure 4. Please report the inference time for SynFormer in the figure/table.
>
> We have updated Figure 4 to additionally include the inference time of SynFormer [A]. When running SynFormer, we increased the beam size to 96 and exhaustiveness to 256 (the default setting in the original paper was a beam width of 24 and exhaustiveness of 64), and ran multiple cycles to match the sampling process of ReaSyn. As shown in the figure, SynFormer falls short of ReaSyn's performance (BU+TD+EB) by a large margin. Moreover, SynFormer performs worse than ReaSyn, which uses only a single direction (BU or TD), demonstrating that ReaSyn outperforms SynFormer through both (1) novel unification of bottom-up and top-down bidirectional pathway prediction and (2) component-specific contribution.
>
> ---
> > Fairness of comparison. ReaSyn’s gains partly come from using three networks and scaling at inference. Comparing this directly to single-pass baselines is somewhat unfair. The paper should clearly state the main contribution and separate the gains from architecture vs. extra inference compute.
>
> We appreciate bringing this to our attention. We have reported SynFormer’s sampling time in the newly updated Figure 4. Even after we use SynFormer multiple times to construct a multi-pass baseline, we can observe that SynFormer falls short of ReaSyn's performance (BU+TD+EB) by a large margin, demonstrating ReaSyn’s superiority over SynFormer as an effective synthesizable drug discovery approach.
>
> We emphasize that ReaSyn uses two networks (autoregressive and Edit Bridge) instead of three. One of the main contributions is that **ReaSyn unifies bottom-up and top-down pathway prediction using a single network** (lines 130-131, 243-264). The autoregressive model of ReaSyn is trained with both bottom-up pathways and top-down pathways, and is able to control the sampling direction by controlling the first token during inference. Compared with SynFormer [A] which has 230M parameters, ReaSyn has a total of 340M parameters (166M for the autoregressive model and 174M for the Edit Bridge model), which is 1.5x bigger.
>
> ---
> *References*
>
> [A] Gao et al., Generative AI for Navigating Synthesizable Chemical Space, PNAS, 2025.
>
> ---
> We hope our response addresses your questions and clarifies any confusion. If our response is satisfactory, we would like to kindly ask you to consider raising your score. Otherwise, we will be happy to further discuss and update the paper.

---

### Official Review · Reviewer_gV5V · 2025-11-09

**Soundness:** 4
**Presentation:** 3
**Contribution:** 3
**Rating:** 4
**Confidence:** 5

**Summary:**

The paper uses an autoregressive models to construct synthesis trees for molecules on bottom up or top down order. this is then used to perform molecular design effectively.

**Strengths:**

- strong empirical results, extensive validation
- interesting use of edit bridge model
- great use of BU and TD combined model

**Weaknesses:**

1) The edit bridge model should be explained in more detail, both in theory and also how practically the implementation is done.

2) Typo: In Section 4 I think the building block space should be larger than the reaction space?

3) The contribution of the paper is methodically good, but unfortunately, the paper currently ignores most of the pioneering work in the area. This needs to be rectified before a higher score can be assigned.

Missing citations to prior work:

Autoregressive Chemical Language Models that process and generate SMILES were introduced in 2017 in
https://arxiv.org/abs/1701.01329
Interestingly, this paper also features a discussion around reaction driven design - the field has seems to have gone full circle, but the problems have been known already ~10 years ago :)

Synthesizability Projection was introduced by Bradshaw https://arxiv.org/pdf/2012.11522 see appendix D3

Modern AI-driven Synthesis Planning was introduced in https://www.nature.com/articles/nature25978 (2018) not in 2019.

Most TDC oracles used in the paper were not introduced by TDC but taken from Guacamol https://doi.org/10.1021/acs.jcim.8b00839 originally developed at BenevolentAI - please cite the original work.

for Morgan Fingerprints, I'd suggest to cite https://pubs.acs.org/doi/10.1021/ci100050t as well

**Questions:**

- what is the validity rate for the generated SMILES building blocks?

---

> ### Author Response · Authors · 2025-11-21
>
> We sincerely appreciate your comments that our paper presents strong empirical results, extensive validation, interesting use of the Edit Bridge model, and great use of BU and TD combination. We address your concerns below.
>
> ---
> > The Edit Bridge model should be explained in more detail, both in theory and also how practically the implementation is done.
>
> The proposed Edit Bridge is a sub-class of Edit Flow [A], a new discrete flow model that overcomes the fixed-length problem of discrete diffusion models while preserving their advantage of processing full sequences. While full details are provided in the original paper [A], we have included the details regarding the Edit Flow in **Section 3 and Section B**. Specifically, as explained in lines 182-196, the original Edit Flow defines a Continuous-Time Markov Chain (CTMC) [B] at the full sequence level rather than at the token level. The CTMC transports sequences from a source (e.g., noise) distribution $p(\boldsymbol{p})$ to a target (e.g., data) distribution $q(\boldsymbol{p})$ via edit operations: token insertions, deletions, and substitutions. The distribution of source ($\boldsymbol{p}_0$) and target ($\boldsymbol{p}_1$) sequence pairs is called coupling $\pi(\boldsymbol{p}_0, \boldsymbol{p}_1)$, whose marginals are $p$ and $q$. Edit Flow uses the empty coupling where $\boldsymbol{p}_0$ is an empty sequence or the uniform coupling where p(p0) is uniform over tokens.
>
> As explained in **Section 4.3**, the novelty of our Edit Bridge lies in its coupling. While the original Edit Flow uses the empty coupling where $\boldsymbol{p}_0$ is an empty sequence or the uniform coupling where $p(\boldsymbol{p}_0)$ is a uniform distribution over tokens, Edit Bridge forms a coupling between a sample $\boldsymbol{p}_0$ generated by ReaSyn’s autoregressive model and the target sequence $\boldsymbol{p}_1$. As shown in Table 8, our Edit Bridge coupling shows much higher $\boldsymbol{p}_0$-$\boldsymbol{p}_1$ alignment rate than the empty or uniform couplings (70.6% vs. 0.0% or 2.6%). Consequently, it requires much fewer edit operations to convert $\boldsymbol{p}_0$ to $\boldsymbol{p}_1$ than the two couplings, resulting in much fewer sampling steps during inference (30.0 vs. 94.6 or 142.9). We have also provided examples of the holistic editing using Edit Bridge in Figures 8 and 9. After ReaSyn’s bidirectional autoregressive model generates a synthetic pathway, ReaSyn’s Edit Bridge model comprehensively considers the entire pathway to further refine it. We are happy to provide additional details if required and update the paper accordingly.
>
> ---
> > Typo: In Section 4 I think the building block space should be larger than the reaction space?
>
> We apologize for the typo. This has been corrected on line 203 of the revised paper (highlighted in green).
>
> ---
> > The contribution of the paper is methodically good, but unfortunately, the paper currently ignores many citations [B-F].
>
> We sincerely appreciate your suggestions regarding the citations and have incorporated all of them into the revised paper (highlighted in green) as below. We believe this has further strengthened our work.
> - Segler et al. [B] in line 217.
> - Bradshaw et al. [C] was already cited in lines 51, 74, 95, and 145. The work was also used as the baselines (DoG-Gen and DoG-AE) in Table 3.
> - Segler et al. [D] in line 40.
> - Brown et al. [E] in line 421.
> - Rogers et al. [F] in line 351.
>
> We are happy to expand the discussion of these works should the reviewer suggest it.
>
> ---
> > What is the validity rate for the generated SMILES building blocks?
>
> We additionally report success rates below. Here, the **success rate** refers to the validity of the entire pathway, requiring the validity of all building block SMILES included in the pathway and compatibility between building blocks and reaction steps. As shown in the table, ReaSyn shows a near-perfect success rate across test sets.
>
> **Table: Synthesizable molecule reconstruction results.**
> | Dataset | Success rate (%) | Reconstruction rate (%) |
> | --- | --- | --- |
> | Enamine | 99.9 | 95.0 |
> | ChEMBL | 99.6 | 31.7 |
> | ZINC250k | 99.9 | 87.9 |
>
> ---
> *References*
>
> [A] Havasi et al., Edit flows: Flow matching with edit operations, arXiv, 2025.
>
> [B] Segler et al. Generating focused molecule libraries for drug discovery with recurrent neural networks, ACS central science, 2018.
>
> [C] Bradshaw et al., Barking up the right tree: an approach to search over molecule synthesis dags, NeurIPS, 2020.
>
> [D] Segler et al., Planning chemical syntheses with deep neural networks and symbolic AI, Nature, 2018.
>
> [E] Brown et al., GuacaMol: benchmarking models for de novo molecular design, Journal of chemical information and modeling, 2019.
>
> [F] Rogers et al., Extended-connectivity fingerprints, Journal of chemical information and modeling, 2010.

---

> > ### Comment · Reviewer_gV5V · 2025-11-21
> > **thank you**
> >
> > thank you, i will raise my score. looking forward to trying out the model!
> >
> > one more tiny question: regarding the smiles validity, if it is so high, why do you then need the lookup based on Levenshtein similarity for building blocks that are invalid smiles? does it help during training somehow?

---

> ### Author Response · Authors · 2025-11-21
>
> We hope our response addresses your questions and clarifies any confusion. If our response is satisfactory, we would like to kindly ask you to consider raising your score. Otherwise, we will be happy to further discuss and update the paper.

---

> ### Author Response · Authors · 2025-11-22
>
> We sincerely appreciate you raising the score. We plan to release the codebase soon for the community.
>
> ---
> > Regarding the SMILES validity, if it is so high, why do you then need the lookup based on Levenshtein similarity for building blocks that are invalid smiles?
>
> The success rate in the previous response was calculated using the Levenshtein distance for building block SMILES. We additionally report **building block validity before applying the Leveshtein distance lookup** below. While ReaSyn’s autoregressive model mostly generates valid SMILES, ReaSyn’s Edit Bridge model is based on edit operations on tokens, which sometimes generates invalid SMILES. Nevertheless, ReaSyn exhibits near-perfect building block validity overall.
>
> There are two reasons for the lookup based on the Levenshtein distance: (1) to increase the success rate by correcting a small number of invalid building blocks into valid building blocks, and (2) to ensure that all generated building blocks belong to the predefined building block set $\mathcal{B}$.
>
> **Table: Synthesizable molecule reconstruction results** on Enamine. The search width, exhaustiveness, number of cycles, and number of Edit Bridge samples in each cycle were set to 1.
> | Step | Building block validity (%) |
> | --- | --- |
> | Autoregressive | 100.0 |
> | Edit Bridge | 88.7 |
> | **Total** | **96.8** |

---

> ### Author Response · Authors · 2025-11-30
>
> We're glad to hear we addressed your concerns, and thank you for raising the score to positive (8) on November 22nd. However, it appears the score has reverted back to the initial score (4). Could you please check again and let us know if you have any further questions? We will be happy to discuss further.

---

### Meta-Review · Area_Chair_MdKo · 2025-12-24

**Summary:**

This paper introduces ReaSyn, an autoregressive model for bottom-up and top-down traversal/generation of synthesis trees for molecular generation. ReaSyn also proposes an editing mechanism (based on a discrete flow formulation) to further improve exploration of the space of synthesizable molecules. Experimental results demonstrate strong performance in goal-directed molecular generation while achieving high synthesizability.

The paper is overall well received and all reviewers highlight the strong experimental results. Most reviewer concerns, like missing references, or missing clarifications about method details, have been successfully addressed during the author rebuttal. The AC believes that all reviewers that originally voted for (weak) rejection, would have updated their scores to support acceptance of this paper. Given the overall positive reception, strong experimental results, timeliness of the topic, and overall high quality, the paper meets the bar for an Oral presentation at the conference.

**Reviewer Concerns:**

Most reviewer concerns related to clarification about the method, its compute usage, baseline comparisons, and additional related work. All concerns were sufficiently addressed in the rebuttal.

**Reviewer Scores:**

Reviewer gV5V: mentioned in discussion that they increased their score (but not by how much). The authors later comment that the score was increased to 8, but the AC will have to disregard this comment as this information was rolled back prior to re-assignment to new area chairs. We will go with the assumption that the reviewer has increased their score into the weak accept or beyond stage, i.e. supports acceptance of this paper.

The weak rejecting reviewer would have likely updated their score to weak accept, given that their concerns were mostly addressed. Both accept voting reviewers would have likely not updated their score — reviewer 5E2P also highlights this in their reviewer response to the rebuttal.

---

### Decision · Program_Chairs · 2026-01-26

Accept (Oral)